# Dem-HEC: High-Entropy Contrastive Fine-Tuning for Countering Natural Corruptions

## Abstract

Neural networks are highly susceptible to natural image corruptions such as noise, blur, and weather distortions, limiting their reliability in real-world deployment. The prime reason to maintain the high integrity against natural corruptions is that these distortions are the primary force of distribution shift intentionally (compression) or unintentionally (blur or weather artifacts). For the first time, through this work, we observe that such corruptions often collapse the network's internal feature space into a high-entropy state, causing predictions to rely on a small subset of fragile features. Inspired by this, we propose a simple yet effective entropy-guided fine-tuning framework, Dem-HEC, that strengthens corruption robustness while maintaining clean accuracy. Our method generates high-entropy samples within a bounded perturbation region to simulate corruption-induced uncertainty and aligns them with clean embeddings using a contrastive loss. In parallel, cross-entropy on both clean and high-entropy samples, combined with knowledge distillation from a teacher snapshot, ensures stable predictions. Dem-HEC is evaluated with numerous neural networks trained on multiple benchmark datasets, demonstrating consistent gains across diverse corruption types and their severities (noise strength), with strong transferability across backbones, including CNNs and Transformers. Our approach highlights entropy regularisation as a scalable pathway to bridging the gap between clean accuracy and real-world robustness.

## 1 Introduction

In this current era of the digital world and high computing, the tremendous success of deep learning models trained end-to-end has led to their deployment in almost every field of vision and on almost every possible digital device, ranging from laptops to mobile devices. However, still contrary to human vision, these systems are still imperfect in handling out-of-distribution (OOD) samples, especially where the samples are affected by natural, also known as common, corruptions (Recht et al., 1806; Azulay & Weiss, 2024; Mitra et al., 2024; Hendrycks & Dietterich, 2019; Pedraza et al., 2022; Agarwal et al., 2024). This kind of robustness against OOD images affected by natural corruption is a crucial objective for machine learning and computer vision tasks, in case they truly need to be autonomous. In general, imaging accuracy is measured as in-distribution performance, which means a model trained and applied to the same kind of data without any distributional shift. But, in practice, deep neural networks (DNNs) mostly observe different data distributions due to an unconstrained environment than what is encountered during training. Surprisingly, modelling every form of common corruption is not feasible, and even including them in training can lead to a significant increase in the computational cost. Therefore, we believe. The robustness must be an inherent part of any network training, because the deployment of models must not be restricted to any environment. For example, the significant number of steps involved in image acquisition introduces several noises in the images. For example, CMOS sensors are prone to several types of noise, including photon shot noise and amplifier noise, particularly in low-light settings (Bigas et al., 2006). Similarly, transferring or storing images on edge-devices requires the use of compression, which itself generates image artifacts. Moreover, if the use of the model is truly universal and ensures that no geographical boundary exists, they have to tackle several environmental factors, such as snow and frost.

**Corruption robustness.** Let $C$ denote a set of corruption functions and $f : X \rightarrow Y$ be a classifier trained on samples from a distribution $D$ that does not include any corruptions from $C$. The robustness of $f$ is evaluated by its average performance when classifying corrupted inputs, where the

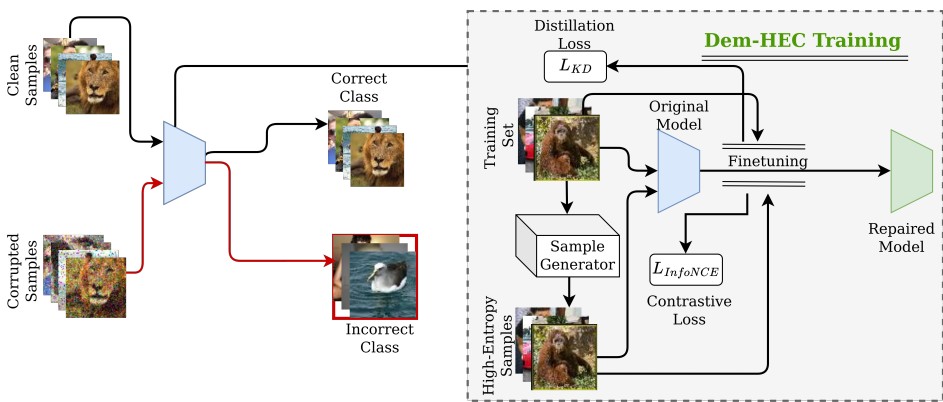

Figure 1: An overview of our Dem-HEC framework. The left panel illustrates how an original model fails to handle the corrupted samples. The right panel details the training procedure, which combines contrastive learning and knowledge distillation to improve robustness.

corruptions are drawn from $C$ (Hendrycks & Dietterich, 2019). Formally, this is expressed as

$$\mathbb{E}_{c \sim C} \, \mathbb{P}_{(x,y) \sim D} \big( f(c(x)) = y \big).$$

Deep neural networks perform worse under such distribution shifts where the training data is different than the testing data (Zhou et al., 2024; Kumar & Agarwal, 2023; Kumar et al., 2025). Before implementing DNNs in the unpredictable and noisy real world, it is essential to assess the consequences of incorrect decisions made by these networks, regardless of the cause, such as image corruption. For robustness of the DNNs, including state-of-the-art transformer, a model trained on clean images suffers on noisy images even if the severity of the noisy data is low (i.e., severity is 1) and further gets worse if severity increases in the ranges of 1 to 5 (Kumar et al., 2025). Similar performance degradation has been noticed for different natural distribution-shifts (Knoll et al., 2019; Darestani et al., 2021).

Extensive research has benchmarked this vulnerability, revealing that different model architectures exhibit unique sensitivities. For instance, Vision Transformers (ViTs) may be robust to noise but susceptible to environmental corruptions, while Convolutional Neural Networks (CNNs) can show the opposite behavior. This indicates that there is no single "silver bullet" architecture that is universally robust, highlighting the need for methods that can bolster a model's resilience regardless of its design. Democratic Training (Sun et al., 2025), defend against Universal Adversarial Perturbations (UAPs). The key insight of this work is that UAPs cause an abnormal decrease in the entropy of a network's hidden layer activations, suggesting that a few dominant features hijack the decision-making process. Consequently, democratic training fine-tunes a model on synthetically generated low-entropy samples to force a more distributed, or "democratic," feature representation.

Inspired by this entropy-based analysis, we address the distinct challenge of robustness against natural corruptions. We hypothesize that, unlike UAPs, which induce feature dominance and low entropy, natural corruptions introduce ambiguity and uncertainty, which can be modeled by an increase in feature space entropy discussed in subsection 3.2. Therefore, we propose a novel fine-tuning framework, Dem-HEC (Democratic High-Entropy samples for Corruption robustness), as described in Figure 1 that takes the opposite approach to Democratic Training Sun et al., 2025. Instead of suppressing dominant features, Dem-HEC encourages the model to learn invariant representations by training it on challenging high-entropy samples. These samples are generated via gradient ascent on the entropy of the model's feature space, pushing the model to make stable predictions even when feature activations are maximally uncertain. To achieve this, we introduce a composite loss function that combines four key objectives: (1) standard cross-entropy on clean images to maintain baseline accuracy, (2) cross-entropy on our generated high-entropy samples to learn robust features, (3) a contrastive loss to ensure that the representations of clean images and their high-entropy counterparts remain semantically similar, and (4) knowledge distillation to prevent the model from catastrophically forgetting the knowledge of the original pre-trained network. We demonstrate through extensive experiments on CIFAR10, CIFAR100, and Tiny-ImageNet with various backbones (ResNet, ViT) that Dem-HEC significantly enhances robustness against a wide range of common corruptions and severities, often outperforming models trained on clean data alone.

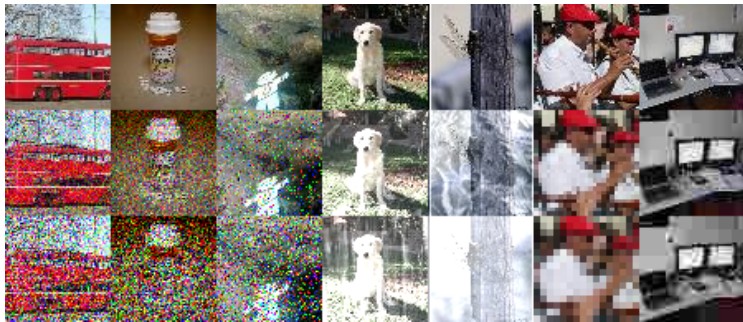

Figure 2: Visual examples of the seven common corruptions used in our evaluation. The first row displays the original clean images. The second and third rows show the corresponding corrupted images at severity levels 3 and 5, respectively. The corruptions, from left to right, are: Gaussian noise, shot noise, impulse noise, snow, frost, pixelate, and JPEG compression.

## 2 NOTATION AND DEFINITIONS

### 2.1 COMMON CORRUPTION

In this work, we focus on seven widely recognized common corruption types that reflect real-world degradations frequently encountered in image acquisition, transmission, and storage. The first category consists of additive noise corruptions: *Gaussian noise*, *Shot noise*, and *Impulse noise*. The second category involves environmental corruptions: *Snow corruption* and *Frost corruption*. Finally, we consider digital corruptions, which are consequences of post-capture transformations: *Pixelation* and *JPEG compression*. Together, these seven corruption types cover a broad range of sensor-level, environmental, and digital artifacts, providing a comprehensive testbed for evaluating the corruption robustness of deep neural networks. Moreover, for comprehensiveness, each corruption has been applied with multiple severities reflecting mild (S1), medium (S3), and high (S5) severity. The corresponding severity parameter has been inspired by the work of Hendrycks & Dietterich, 2019 and is given at[1]. Figure 2 shows the challenge that the proposed research is handling by tackling the loss of visual cues at high severities, and the strength of the proposed research. The details about these corruptions are provided in the appendix A.1.

### 2.2 EVALUATION METRICS

**Corrupted Accuracy (CAcc.):** This metric measures the accuracy of corrupted examples (where $y_x$ represents the label of sample $x$):

$$CAcc. = \sum_{x \in X} \frac{|f(x + \delta) = y_x|}{|X|} \tag{1}$$

### 2.3 ENTROPY OF A NEURAL NETWORK

In information theory, Shannon entropy is a fundamental measure that quantifies the average level of uncertainty or information contained in the outcomes of a random variable. First introduced by Claude Shannon (Shannon, 1948), this concept captures how much "surprise" or unpredictability is associated with a probabilistic system. Formally, let $v$ be a random variable that can take values from a set $V$ with an associated probability distribution $p : V \to [0, 1]$. The Shannon entropy of $v$ is expressed as:

$$H(v) = -\sum_{v \in V} p(v) \log p(v), \tag{2}$$

where the summation is taken over all possible outcomes of $v$.

Entropy has been widely adopted in the context of neural networks to characterize the level of uncertainty in their internal representations or predictions. Prior works have proposed different strategies for estimating neural entropy at various levels of abstraction. In this study, we focus

---

[1]https://github.com/bethgelab/imagecorruptions

on computing *layer-wise entropy* to investigate how common corruptions alter the internal feature distributions of a network. A detailed description of this formulation is presented in subsection 3.1.

## 2.4 PROBLEM FORMULATION: COMMON CORRUPTION ROBUSTNESS

Let $F$ denote a neural network classifier obtained from a third party, and let $x \in \mathbb{R}^{H \times W \times C}$ be a clean input with ground-truth label $y$. Consider a family of corruption operators

$$\mathcal{G} = \{g_c(\cdot, s) \mid c \in \mathcal{C}, \ s \in \{1, 3, 5\}\},$$

where each $g_c : \mathbb{R}^{H \times W \times C} \to \mathbb{R}^{H \times W \times C}$ represents a corruption of type $c$ (e.g., Gaussian noise, shot noise, impulse noise, snow, frost, pixelation, JPEG compression) applied with severity level $s$.

The *common corruption robustness problem* is to design a defense strategy such that the network's predictions remain reliable under these corruptions:

$$\arg\max F(x) = y \quad \Longrightarrow \quad \arg\max F(g_c(x, s)) = y, \ \ \forall c \in \mathcal{C}, \ s \in \{1, \ldots, 5\}. \tag{3}$$

At the same time, the defense must preserve the classifier's performance on clean data, i.e., the accuracy on uncorrupted inputs $x$ should remain close to that of the original network.

## 3 OUR APPROACH

To investigate how natural corruptions affect model behavior, we conduct a systematic analysis through the lens of entropy. Specifically, we examine the *layer-wise entropy* of a given network when processing both clean and corrupted inputs. As we demonstrate in Section 3.2, the presence of natural corruptions such as Gaussian noise, shot noise, or JPEG compression often increases entropy compared to clean data, and this addition becomes increasingly pronounced at deeper layers. Motivated by these findings, we propose **Dem-HEC**, an entropy-guided training framework that enhances model robustness against natural corruptions by encouraging balanced feature representations.

## 3.1 ENTROPY MEASUREMENT

We begin by defining how entropy is measured in our setting. Consider a neural network $F$ consisting of $n$ layers. Each layer $l$ can be treated as a random variable characterized by its input $x_l$ and output $x_{l+1}$. For a layer with $d_l$ neurons, given input

$$x_l = \{x_l^0, x_l^1, \ldots, x_l^{d_l - 1}\},$$

its activations are computed as

$$\chi_l = \sigma(W_l x_l + b_l),$$

where $W_l$ and $b_l$ denote the weights and biases of layer $l$, and $\sigma(\cdot)$ is its activation function. The normalized activation distribution is obtained via

$$p_l = \mathrm{softmax}(\chi_l).$$

Finally, the *layer-wise entropy* is defined as

$$H_l = -\sum_{k=0}^{d_l - 1} p_l(k) \log p_l(k). \tag{4}$$

Intuitively, we treat the activation probability $p_l(k)$ of neuron $k$ as the likelihood of it being active, and compute the Shannon entropy over all neurons. A higher entropy $H_l$ indicates greater uncertainty or feature diversity, while lower entropy reflects higher certainty or dominance of a small subset of neurons. Under natural corruptions, we often observe an abnormal increase in entropy, suggesting that corrupted inputs cause the network to overly rely on spurious features rather than balanced feature representations.

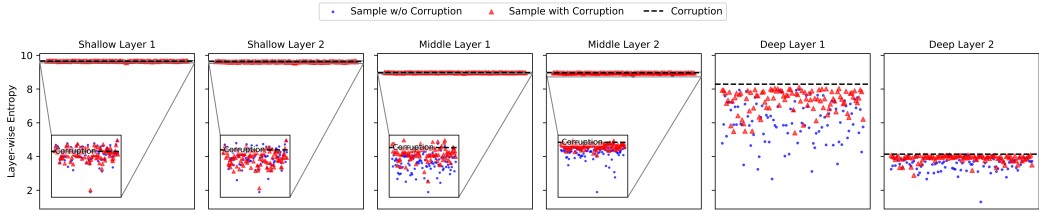

Figure 3: Layer-wise entropy for a ResNet-20 on CIFAR10 with pixelation corruption with severity 5. Entropy clearly separates clean (blue) and corrupted (red) samples in deep layers, while remaining uniformly high for both in shallow and middle layers.

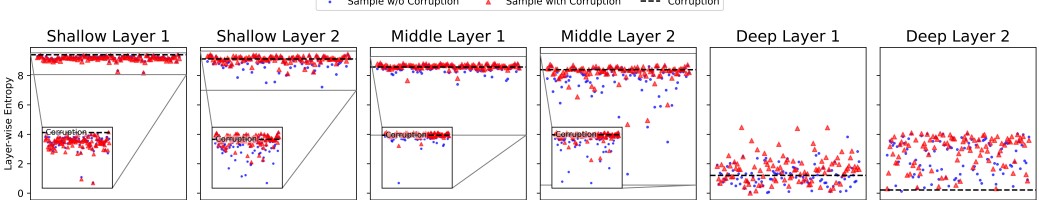

Figure 4: Layer-wise entropy for repaired ResNet-20 using Dem-HEC on CIFAR10 with pixelation corruption with severity 5. Entropy clearly separates clean (blue) and corrupted (red) samples in deep layers, while remaining uniformly high for both in shallow and middle layers.

## 3.2 ENTROPY ANALYSIS

To understand how natural corruptions influence the behavior of a trained neural network, we conduct an empirical study on the *layer-wise entropy* of the model as follows: **Step 1.** Given a pretrained neural network, we collect a set of clean test samples and compute their layer-wise entropy as defined in Equation (4). **Step 2.** Apply different natural corruptions (e.g., Gaussian noise, shot noise, impulse noise, snow, frost, pixelation, JPEG compression) with varying severity levels to the same set of samples. **Step 3.** Compute and compare the layer-wise entropy of clean inputs versus corrupted inputs. **Step 4.** Analyze the evolution of entropy across shallow, middle, and deep layers to understand how corruptions alter uncertainty.

As illustrated in Figure 3, before applying the proposed Dem-HEC, at shallow layers, the entropy distributions of clean and corrupted inputs are close to each other, indicating that early convolutional features are relatively stable. However, as inputs propagate through middle and deeper layers, corrupted samples consistently exhibit **higher entropy** than their clean counterparts. This effect becomes more pronounced at deeper layers, where natural corruptions induce substantial ambiguity in the learned representations. Figure 4 shows the entropy distribution after training with our high-entropy samples, where clean and corrupted inputs now fall within the same entropy range across all layers. This demonstrates that the generated high-entropy samples successfully reproduce the feature-space uncertainty patterns induced by real natural corruptions, while still preserving semantic structure. These results confirm that the high-entropy samples used in Dem-HEC are consistent with true corruption behavior and effectively guide the model toward stable, corruption-robust representations.

These results suggest that, unlike UAPs, which inject dominant features and reduce entropy, natural corruptions increase entropy by dispersing feature activations, thereby making the model less confident about its predictions. In other words, corruptions distort discriminative cues, forcing the network to rely on noisy or occluded signals, which leads to higher uncertainty. Our analysis thus highlights a key contrast: *UAPs enforce artificial certainty (low entropy), while natural corruptions degrade representation quality and amplify uncertainty (high entropy)*.

## 3.3 PROPOSED DEM-HEC

To mitigate the effect of natural corruptions on neural networks, we propose Dem-HEC, a general framework applicable to different architectures (e.g., CNNs such as ResNet-18/56, RepVGG-A0/A2, or Transformers such as ViT) and datasets (CIFAR10, CIFAR100, Tiny ImageNet). Unlike existing defenses designed for Universal Adversarial Perturbations (UAPs), which focus on reducing

---

**Algorithm 1** Dem-HEC Training (architecture- and dataset-agnostic)

---

**Require:** Pretrained model $f(\cdot; \theta)$; teacher copy $f(\cdot; \theta_T)$ (frozen); hyperparameters $\alpha$, $\lambda_C$, $\lambda_{\mathrm{KD}}$, temperature $T$; PGA steps $T_{\mathrm{he}}$, step size $\eta$, radius $\epsilon$.

1: **for** epoch $= 1, \ldots, E$ **do**
2:    **for** minibatch $\mathcal{B} = \{(x_i, y_i)\}_{i=1}^{B}$ **do**
3:       **High-entropy samples:** for each $x_i$, compute $x_i' \leftarrow$ HE_GENERATE$(x_i; \epsilon, \eta, T_{\mathrm{he}})$
4:       **Forward:** obtain logits $z_i = f(x_i; \theta)$ and $z_i' = f(x_i'; \theta)$
5:       **Embeddings:** $\boldsymbol{v}_i = \mathrm{norm}(g(x_i; \theta))$,   $\boldsymbol{v}_i' = \mathrm{norm}(g(x_i'; \theta))$
6:       **Teacher logits (clean):** $z_i^{(T)} = f(x_i; \theta_T)$
7:       **Losses:**

$$\mathcal{L}_{\mathrm{CE}}^{\mathrm{clean}} = \tfrac{1}{B} \sum_i -\log \mathrm{softmax}(z_i)[y_i],$$

$$\mathcal{L}_{\mathrm{CE}}^{\mathrm{he}} = \tfrac{1}{B} \sum_i -\log \mathrm{softmax}(z_i')[y_i],$$

$$\mathcal{L}_{\mathrm{InfoNCE}} \text{ from } \{\boldsymbol{v}_i\}, \{\boldsymbol{v}_i'\},$$

$$\mathcal{L}_{\mathrm{KD}} = \tfrac{T^2}{B} \sum_i \mathrm{KL}\big(\sigma(z_i/T) \,\|\, \sigma(z_i^{(T)}/T)\big)$$

8:       **Total loss:**

$$\mathcal{L}_{\mathrm{total}} = (1 - \alpha)\, \mathcal{L}_{\mathrm{CE}}^{\mathrm{clean}} + \alpha\, \mathcal{L}_{\mathrm{CE}}^{\mathrm{he}} + \lambda_C\, \mathcal{L}_{\mathrm{InfoNCE}} + \lambda_{\mathrm{KD}}\, \mathcal{L}_{\mathrm{KD}}.$$

9:       **Update:** $\theta \leftarrow \theta - \eta_{\mathrm{opt}} \nabla_\theta \mathcal{L}_{\mathrm{total}}$
10:    **end for**
11: **end for**

---

**Algorithm 2** HIGH-ENTROPY SAMPLE GENERATOR

---

**Require:** Input $x$, radius $\epsilon$, step size $\eta$, steps $T_{\mathrm{he}}$

1: Initialize $x^{(0)} \leftarrow \mathrm{clip}\big(x + \mathcal{U}(-\epsilon, \epsilon)\big)$    (optional random start)
2: **for** $t = 0$ to $T_{\mathrm{he}} - 1$ **do**
3:    Compute gradient $\boldsymbol{g}^{(t)} \leftarrow \nabla_{x^{(t)}} H\big(\mathrm{softmax}\big(f(x^{(t)})\big)\big)$
4:    Ascent step $x^{(t+1)} \leftarrow x^{(t)} + \eta \cdot \mathrm{sign}\big(\boldsymbol{g}^{(t)}\big)$
5:    Project $x^{(t+1)} \leftarrow \Pi_{\mathcal{B}_\epsilon(x)}\big(x^{(t+1)}\big)$ and clip to $[0, 1]$
6: **end for**
7: **return** $x' \leftarrow x^{(T_{\mathrm{he}})}$

---

overconfident low-entropy activations, our method explicitly accounts for the opposite phenomenon: natural corruptions tend to induce high-entropy predictions (greater uncertainty). Dem-HEC therefore regularizes networks to handle corrupted high-entropy samples while maintaining strong accuracy on clean data.

### 3.3.1 BACKBONE AND PROBLEM SETUP

Let $f(\cdot; \theta)$ be a pretrained classifier with parameters $\theta$. Given an input image $x \in \mathbb{R}^{H \times W \times C}$ and label $y \in \{1, \ldots, K\}$, the model produces logits $z = f(x; \theta)$ and predictive distribution

$$p(y \mid x) = \mathrm{softmax}(z). \tag{5}$$

The standard cross-entropy loss is

$$\mathcal{L}_{\mathrm{CE}}\big(f(x; \theta), y\big) = -\log p(y \mid x). \tag{6}$$

We adopt *partial fine-tuning* (freeze early layers, update higher blocks and head) to retain general features while adapting to corruption robustness.

### 3.3.2 HIGH-ENTROPY SAMPLE GENERATION

Natural corruptions typically increase predictive uncertainty in deep layers. We simulate this training signal by synthesizing a *high-entropy* variant $x'$ of $x$ via constrained entropy maximization shown in Algorithm 2.

Let the Shannon entropy of the model output be

$$H(p(\cdot \mid x)) = -\sum_{k=1}^{K} p_k(x) \log\big(p_k(x) + \varepsilon_0\big), \tag{7}$$

with a small $\varepsilon_0 > 0$ for numerical stability. We solve

$$x' = \arg \max_{\|x'-x\|_\infty \leq \epsilon} H(p(\cdot \mid x')), \tag{8}$$

using $T$ steps of Projected Gradient Ascent (PGA):

$$x^{(t+1)} = \Pi_{\mathcal{B}_\epsilon(x)}\Big(x^{(t)} + \eta \cdot \mathrm{sign}\Big(\nabla_{x^{(t)}} H\Big(p(\cdot \mid x^{(t)})\Big)\Big)\Big), \tag{9}$$

where $\eta$ is the step size and $\Pi_{\mathcal{B}_\epsilon(x)}$ projects onto the $\ell_\infty$ ball of radius $\epsilon$ around $x$ (and to the valid pixel range).

### 3.3.3 CONTRASTIVE REPRESENTATION ALIGNMENT

Let $g(\cdot; \theta)$ be a representation extractor (e.g., penultimate layer), and define $\boldsymbol{v} = \mathrm{norm}\big(g(x; \theta)\big)$ and $\boldsymbol{v}' = \mathrm{norm}\big(g(x'; \theta)\big)$, with $\mathrm{norm}(\cdot)$ denoting $\ell_2$-normalization. For a batch of size $B$, $\{\boldsymbol{v}_i\}_{i=1}^{B}$ and $\{\boldsymbol{v}'_i\}_{i=1}^{B}$, the symmetric InfoNCE loss is

$$\mathcal{L}_{\mathrm{InfoNCE}} = -\frac{1}{2B} \sum_{i=1}^{B} \left[ \log \frac{\exp\big(\mathrm{sim}(\boldsymbol{v}_i, \boldsymbol{v}'_i)/\tau\big)}{\sum_{j=1}^{B} \exp\big(\mathrm{sim}(\boldsymbol{v}_i, \boldsymbol{v}'_j)/\tau\big)} \right.$$
$$\left. + \log \frac{\exp\big(\mathrm{sim}(\boldsymbol{v}'_i, \boldsymbol{v}_i)/\tau\big)}{\sum_{j=1}^{B} \exp\big(\mathrm{sim}(\boldsymbol{v}'_i, \boldsymbol{v}_j)/\tau\big)} \right], \tag{10}$$

where $\mathrm{sim}(\boldsymbol{u}, \boldsymbol{v}) = \boldsymbol{u}^\top \boldsymbol{v}$ and $\tau > 0$ is a temperature.

### 3.3.4 KNOWLEDGE DISTILLATION FOR CLEAN-DATA STABILITY

To avoid forgetting on clean inputs, we distill from a frozen teacher $f(\cdot; \theta_T)$ into the student $f(\cdot; \theta_S)$ using softened logits:

$$\mathcal{L}_{\mathrm{KD}} = T^2 \cdot \mathrm{KL}\big(\sigma\big(z_S/T\big) \,\|\, \sigma\big(z_T/T\big)\big), \tag{11}$$

where $z_S = f(x; \theta_S)$, $z_T = f(x; \theta_T)$, $\sigma$ is softmax, and $T > 0$ is the distillation temperature.

### 3.3.5 TOTAL OBJECTIVE

The complete Dem-HEC loss (per minibatch) combines clean and high-entropy CE terms, contrastive alignment, and KD follows Algorithm 1:

$$\mathcal{L}_{\mathrm{total}} = (1-\alpha)\,\mathcal{L}_{\mathrm{CE}}(x, y) + \alpha\,\mathcal{L}_{\mathrm{CE}}(x', y) + \lambda_C\,\mathcal{L}_{\mathrm{InfoNCE}} + \lambda_{\mathrm{KD}}\,\mathcal{L}_{\mathrm{KD}}, \tag{12}$$

with trade-off coefficients $\alpha \in [0, 1]$, $\lambda_C \geq 0$, and $\lambda_{\mathrm{KD}} \geq 0$.

## 4 EXPERIMENTAL SETUP

### 4.1 DATASETS AND MODELS

In our experiments, we evaluate the proposed Dem-HEC framework on three widely used benchmark datasets: CIFAR10 (Krizhevsky, 2009), CIFAR100 (Krizhevsky, 2009), and Tiny-ImageNet (or referred to as ImageNet200). To assess our method across a range of model complexities, we select architectures with diverse parameter counts. For CIFAR10 and CIFAR100, we adopt four pretrained

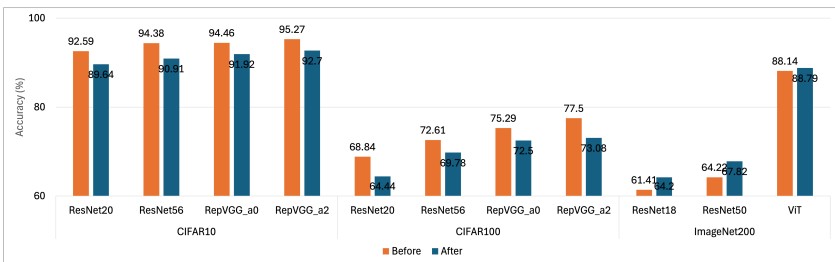

Figure 5: Clean accuracy of models on CIFAR10, CIFAR100, and Tiny-ImageNet (or ImageNet200). The comparison shows performance before and after applying Dem-HEC, illustrating that the original accuracy on uncorrupted data is maintained across all architectures. While on a small scale, a marginal drop has been noticed, on large resolution images, the proposed approach improves the performance on the clean images.

Table 1: Corruption Accuracy (CAcc.) on CIFAR10-C, comparing performance before and after applying Dem-HEC. Our method yields significant robustness gains across all models, particularly for noise-based corruptions and at higher severity levels (S3 and S5).

| Backbone | ResNet20 | | | | | | ResNet56 | | | | | |
|---|---|---|---|---|---|---|---|---|---|---|---|---|
| Severity | S1 | | S3 | | S5 | | S1 | | S3 | | S5 | |
| Corruption | Before | After | Before | After | Before | After | Before | After | Before | After | Before | After |
| Gaussian | 71.37 | **88.16** | 30.17 | **74.18** | 21.23 | **61.47** | 75.71 | **89.98** | 37.02 | **78.04** | 25.83 | **66.97** |
| Shot | 80.92 | **88.90** | 43.46 | **79.16** | 25.86 | **64.00** | 83.98 | **90.60** | 50.49 | **82.44** | 31.37 | **68.47** |
| Impulse | 79.90 | **86.09** | 58.16 | **73.65** | 22.95 | **39.82** | 83.17 | **87.00** | 60.66 | 74.71 | 22.70 | **43.35** |
| Snow | 85.58 | **86.94** | 77.12 | **80.08** | 68.33 | **75.81** | 88.02 | **88.64** | 81.20 | **82.59** | 74.35 | **78.39** |
| Frost | 86.59 | **87.62** | 68.99 | **78.36** | 55.55 | **70.86** | 89.35 | 89.22 | 74.84 | **81.92** | 62.33 | **75.61** |
| Pixelate | 88.89 | 88.72 | 74.97 | **86.59** | 39.85 | **73.29** | 91.53 | 90.09 | 80.40 | **88.87** | 44.73 | **78.53** |
| JPEG | 82.94 | **87.72** | 74.88 | **85.29** | 68.28 | **83.29** | 85.25 | **89.32** | 77.28 | **87.12** | 71.20 | **85.43** |
| Backbone | RepVGG_a0 | | | | | | RepVGG_a2 | | | | | |
| Severity | S1 | | S3 | | S5 | | S1 | | S3 | | S5 | |
| Corruption | Before | After | Before | After | Before | After | Before | After | Before | After | Before | After |
| Gaussian | 71.93 | **90.99** | 20.99 | **79.69** | 14.37 | **69.29** | 77.22 | **91.51** | 30.34 | **80.95** | 19.04 | **71.42** |
| Shot | 82.89 | **91.56** | 38.23 | **84.17** | 19.30 | **71.40** | 86.52 | **92.36** | 50.15 | **85.40** | 27.65 | **74.01** |
| Impulse | 84.22 | **89.39** | 60.34 | **80.44** | 16.08 | **50.10** | 82.52 | **90.04** | 58.85 | **81.91** | 21.63 | **54.42** |
| Snow | 89.16 | **89.56** | 82.70 | **83.63** | 77.06 | **80.40** | 89.19 | **90.55** | 83.82 | **84.90** | 77.34 | **80.79** |
| Frost | 90.79 | 90.62 | 77.67 | **84.35** | 66.18 | **79.46** | 91.58 | 91.28 | 79.71 | **84.89** | 68.94 | **79.86** |
| Pixelate | 92.78 | 91.14 | 85.86 | **89.19** | 50.31 | **77.95** | 93.20 | 92.04 | 85.58 | **90.16** | 50.27 | **80.54** |
| JPEG | 87.19 | **90.37** | 79.90 | **87.85** | 74.49 | **86.14** | 87.87 | **91.02** | 80.98 | **88.77** | 75.26 | **86.70** |

architectures from (Chen): ResNet-20 (0.27M params), ResNet-56 (0.66M params), RepVGG-A0 (489.08M params), and RepVGG-A2 (1850.1M params). For Tiny-ImageNet, we employ three diverse backbones: ResNet-18, ResNet-50, and a large-scale Vision Transformer (ViT-L) with 304M parameters. This selection of models allows us to test the scalability and generalizability of our method. When applying Dem-HEC, we compute entropy primarily at the final pooling or dense layer, as the impact of common corruptions on layer-wise entropy becomes most pronounced in deeper layers, consistent with the analysis presented in Figure 3. The implementation details are also given in the appendix A.3.

## 5 RESULTS AND ANALYSIS

To validate the effectiveness of our proposed Dem-HEC framework, we conducted a comprehensive evaluation across three benchmark datasets (CIFAR10, CIFAR100, Tiny-ImageNet200) and seven different model architectures. We assess performance on both clean data and data subjected to 7 types of common corruptions at varying severity levels.

### 5.1 PERFORMANCE ON CLEAN DATA

A crucial requirement for any robustness enhancement technique is the preservation of performance on uncorrupted (clean) data. Figure 5 illustrates the clean accuracy of all models before and after applying Dem-HEC. The results show that while on the small-scale datasets (CIFAR), the proposed model exhibits slightly lower performance (in the range 2.5 to 4.4%) than the base models (although not always), but interestingly, better performance on the large-scale dataset (Tiny ImageNet). For

Table 2: Corruption Accuracy (CAcc.) on Tiny-ImageNet-C, comparing performance before and after applying Dem-HEC. Our method yields significant robustness gains across all models, particularly for noise-based corruptions and at higher severity levels (S3 and S5).

| Backbone | ResNet18 | | | | | | ResNet50 | | | | | | ViT | | | | | |
|---|---|---|---|---|---|---|---|---|---|---|---|---|---|---|---|---|---|---|
| Severity | S1 | | S3 | | S5 | | S1 | | S3 | | S5 | | S1 | | S3 | | S5 | |
| Corruption | Before | After | Before | After | Before | After | Before | After | Before | After | Before | After | Before | After | Before | After | Before | After |
| Gaussian | 46.08 | **54.36** | 16.30 | **24.95** | 7.38 | **10.58** | 49.85 | **57.01** | 14.83 | **27.43** | 5.32 | **12.73** | 79.88 | **80.45** | 54.61 | **59.72** | 34.42 | **39.40** |
| Shot | 45.19 | **53.62** | 24.90 | **34.83** | 9.36 | **13.80** | 49.50 | **55.89** | 25.81 | **36.85** | 7.23 | **15.57** | 80.16 | **80.14** | 65.99 | **68.52** | 39.33 | **43.19** |
| Impulse | 45.76 | **50.01** | 20.34 | **27.33** | 5.66 | **6.63** | 49.26 | **52.67** | 19.18 | **32.08** | 4.49 | **9.33** | 78.77 | **78.65** | 63.91 | **62.04** | 31.80 | **28.80** |
| Snow | 42.61 | **47.97** | 27.28 | **33.36** | 17.90 | **22.59** | 44.46 | **51.25** | 28.57 | **36.57** | 18.42 | **27.14** | 78.41 | **78.60** | 67.76 | **69.98** | 57.83 | **62.98** |
| Frost | 41.66 | **47.37** | 31.64 | **38.70** | 22.36 | **28.79** | 43.69 | **50.99** | 31.48 | **41.51** | 21.58 | **31.46** | 79.62 | **79.76** | 71.35 | **72.47** | 61.90 | **63.73** |
| Pixelate | 50.48 | **54.57** | 40.79 | **47.37** | 27.51 | **38.02** | 52.46 | **58.02** | 44.19 | **52.26** | 31.50 | **43.28** | 82.29 | **81.39** | 71.55 | **73.95** | 59.78 | **64.11** |
| JPEG | 48.93 | **53.00** | 47.98 | **51.86** | 43.01 | **47.93** | 51.23 | **57.20** | 50.14 | **55.95** | 44.41 | **51.43** | 80.73 | **79.45** | 78.65 | **77.46** | 71.23 | **72.07** |

instance, the ResNet56 model on CIFAR100 sees a decrease from 72.61% to 69.78%, while the approximately similar-sized network, i.e., ResNet50, sees a jump from 64.22% to 67.82 % on the ImageNet200 dataset. The robustness of the network in handling large-scale datasets demonstrates that the proposed approach is scalable and can handle the complexity present in high-resolution images better than in low-resolution images.

## 5.2 ROBUSTNESS AGAINST COMMON CORRUPTIONS

We now analyze the core contribution of Dem-HEC: its ability to enhance model resilience against common corruptions. As showcase in the Figure 2, the high severity noise completly destroy the image features; therefore, robutsness in handling such vast environmental corruption can reflect the genuine strength of the proposed approach. The jump of up to 54% (RepVGG a0 on CIFAR10) shows that the proposed approach can achieve such a feat; the discussion is provided further.

### 5.2.1 ROBUSTNESS ON CIFAR10-C AND CIFAR100-C

As shown in Table 1 and Table 5 (appendix), applying Dem-HEC leads to dramatic improvements in corruption accuracy (CAcc) across all four architectures tested on CIFAR10-C and CIFAR100-C. The most significant gains on CIFAR10-C are observed for high-frequency noise corruptions. For example, the accuracy of RepVGG-A0 under Gaussian noise at the highest severity (S5), from a near-failure rate of 14.37% to 69.29%, a relative increase of over 380%. Similarly, under Shot noise, its accuracy improves from 19.30% to 71.40%. This trend is scalable in handling a large number of classes of CIFAR100-C, where RepVGG-A2's accuracy on Shot noise at severity 5 is more than tripled from 9.49% to 29.28%. A key trend is that the efficacy of Dem-HEC becomes more pronounced as the corruption severity increases. While the baseline models often suffer a catastrophic performance collapse at severity levels 3 and 5, the Dem-HEC-finetuned models exhibit remarkable resilience. For instance, on CIFAR100-C, the ResNet-20 improves its accuracy on JPEG compression artifacts at S5 from 33.90% to 52.65%. Even for corruptions where the baseline is relatively strong, such as Snow, Dem-HEC consistently provides a performance lift, pushing the ResNet-56 accuracy from 74.35% to 78.39% at S5 on CIFAR10-C. This consistent improvement across diverse models and corruption types validates our hypothesis that encouraging high-entropy, distributed feature representations is a generalizable defense against corruption-induced performance degradation.

### 5.2.2 SCALABILITY AND PERFORMANCE ON TINY-IMAGENET200-C

The experiment, detailed in Table 2, tests the scalability of Dem-HEC on both CNN and Transformer architectures on Tiny-ImageNet200-C, which features 200 classes and higher-resolution images. For ResNet-18 and ResNet-50, Dem-HEC continues to provide significant robustness gains, boosting ResNet-50's accuracy on Frost corruption at severity 5 (S5) from 21.58% to 31.46%. The analysis on the ViT model, an inherently more robust architecture, offers a nuanced insight. While the performance gains from Dem-HEC are more modest compared to CNNs, our method still enhances its resilience, particularly at high severities for corruptions like Snow (improving from 57.83% to 62.98% at S5). The smaller margin suggests that ViT's self-attention mechanism may already promote a more "democratic" feature representation. Nevertheless, the ability of Dem-HEC to further improve such a strong baseline underscores its value as a versatile, robustness-enhancing tool.

Table 3: Comparison of corruption-wise accuracy across various robustness methods on Tiny ImageNet dataset with ViT backbone. Our Dem-HEC approach shows the highest improvement across all corruption categories.

| Corruption | Pad-Crop | TA | BA | BA (AA) | AugMix | AugMax | IPMix | DV+AP+JSD | Dem-HEC (Ours) |
|---|---|---|---|---|---|---|---|---|---|
| Noise | 20.94 | 25.27 | 21.18 | 24.62 | 29.15 | 30.18 | 28.80 | 34.44 | **60.10** |
| Blur | 17.27 | 31.23 | 18.01 | 28.36 | 30.77 | 31.72 | 28.04 | 37.04 | **62.88** |
| Weather | 13.41 | 22.88 | 13.54 | 20.29 | 19.94 | 20.65 | 20.79 | 29.17 | **70.71** |
| Digital | 20.99 | 33.19 | 21.53 | 33.07 | 32.26 | 33.49 | 32.34 | 40.24 | **67.53** |

Table 4: Robustness comparison across common corruption types. Our method demonstrates consistent improvements over Ranabhat et al. (2025) across all corruption categories on the CIFAR10 dataset with the ResNet20 backbone.

| Method | Clean | Gaussian | Shot | Impulse | Snow | Frost | Pixelate | JPEG |
|---|---|---|---|---|---|---|---|---|
| Ranabhat et al. (2025) | 83.66 | 65.90 | 69.58 | 62.92 | 75.34 | 73.95 | 81.81 | 76.08 |
| Dem-HEC (Ours) | **89.64** | **74.60** | **77.35** | **66.52** | **80.94** | **78.94** | **82.86** | **85.43** |

### 5.3 COMPARISON WITH EXISTING ROBUSTNESS METHODS

Table 3 presents a systematic comparison of robustness performance across four high-level corruption families: Noise (Gaussian, Shot, Impulse), Blur (Motion, Glass, Defocus, Zoom), Weather (Snow, Frost, Fog, Brightness), and Digital (JPEG, Elastic, Pixelate, Contrast). The reported accuracy values represent the mean performance averaged over all individual corruptions and severity levels within each category. We compare Dem-HEC with widely used robustness-enhancing methods, including Pad-Crop, TA Müller & Hutter (2021), BA Hoffer et al. (2020), AA Cubuk et al. (2019), AugMix Hendrycks et al. (2019), AugMax Wang et al. (2021), IPMix Huang et al. (2023), and DV+AP+JSD Kim et al. (2025). On Tiny-ImageNet-C, as shown in Table 3, Dem-HEC exhibits substantial improvements, particularly for high-severity corruptions, underscoring its scalability to large-resolution datasets and transformer backbones.

We now include results against the recent work of Ranabhat et al. (2025), which proposes a multi-scale push–pull mechanism with channel attention for corruption robustness. As shown in Table 4, Dem-HEC consistently outperforms this method across all seven corruptions and also improves clean accuracy, indicating that our entropy-guided learning provides stronger feature stability.

## 6 CONCLUSION

In this work, we addressed the critical vulnerability of deep neural networks to natural corruptions, which we identify as a shift towards high-entropy, uncertain feature representations. We introduced Dem-HEC, a novel fine-tuning framework that directly confronts this issue by training models on synthetically generated high-entropy samples. By combining contrastive representation alignment with dual cross-entropy and knowledge distillation, our method learns to produce stable predictions even when internal features are maximally uncertain. Our extensive evaluations across CIFAR10, CIFAR100, and Tiny-ImageNet demonstrate that Dem-HEC significantly enhances robustness against a wide array of corruptions and severities without compromising performance on clean data (especially on large-scale datasets). The framework's effectiveness across diverse architectures, including both CNNs and Vision Transformers, validates our approach as a scalable and generalizable solution.

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

# A APPENDIX

## A.1 COMMON CORRUPTION

In this work, we focus on seven widely recognized common corruption types that reflect real-world degradations frequently encountered in image acquisition, transmission, and storage. The first category consists of additive noise corruptions: *Gaussian noise*, which is a common disturbance in low-light conditions or faulty sensor environments, modelled as a signal-independent additive noise with a zero-mean Gaussian distribution. *Shot noise*, also referred to as Poisson noise, arises from the discrete nature of photons in optical sensors and is particularly prevalent in low-exposure or high-sensitivity imaging scenarios. *Impulse noise*, the color analogue of salt-and-pepper noise, appears due to bit errors in transmission or malfunctioning pixels in digital sensors, introducing sharp intensity spikes. The second category involves environmental corruptions. *Snow corruption* introduces white, irregular particles across the scene, imitating obstructive precipitation that reduces visibility and alters texture distribution. *Frost corruption* mimics the accumulation of ice crystals on a lens or window surface, producing distortions similar to imaging through frozen glass. Both snow and frost alter the global scene appearance and occlude local details, challenging a model's ability to extract meaningful representations. Finally, we consider digital corruptions, which are consequences of post-capture transformations. *Pixelation* occurs when low-resolution images are upsampled, leading to blocky structures and loss of fine details, a phenomenon frequently observed in digital zoom or low-bandwidth video transmission. *JPEG compression* is a lossy encoding scheme widely used in digital storage and web transmission, where aggressive compression at high ratios introduces block artifacts and loss of high-frequency details. Together, these seven corruption types cover a broad range of sensor-level, environmental, and digital artifacts, providing a comprehensive testbed for evaluating the corruption robustness of deep neural networks. Moreover, for comprehensiveness, each corruption has been applied with multiple severities reflecting mild (S1), medium (S3), and high (S5) severity. The corresponding severity parameter has been inspired by the work of Hendrycks & Dietterich, 2019 and is given at[2]. Figure 2 shows the challenge that the proposed research is handling by tackling the loss of visual cues at high severities, and the strength of the proposed research.

## A.2 ROBUSTNESS SCALABILITY ON HIGH RESOLUTION DATASET

We evaluated Dem-HEC on ImageNette, a 10-class subset of the original ImageNet dataset using AlexNet. Since the image resolution of ImageNette is in sync with that of full ImageNet, the consistently improved performance of Dem-HEC demonstrates its image resolution-agnostic nature in handling image corruption. Robustness of the proposed Dem-HEC on the ImageNet subset with the AlexNet backbone is shown in Table 6. Please note that the proposed approach not only increases corruption robustness but also retains clean accuracy, clearly demonstrating the ideal trade-off between accuracy and robustness. Our preliminary experiments on full ImageNet with a 5% class-balanced subset demonstrate that the proposed Dem-HEC with a ViT backbone preserves accuracy on clean images (with a slight drop of 0.04%) but increases robustness across various corruptions (Noise, Environmental, Compression, Blur, etc.) by 2-3%. It is interesting to note that, despite

---

[2]https://github.com/bethgelab/imagecorruptions

Table 5: Corruption Accuracy (CAcc.) on CIFAR100-C, comparing performance before and after applying Dem-HEC. Our method yields significant robustness gains across all models, particularly for noise-based corruptions and at higher severity levels (S3 and S5).

| Backbone | ResNet20 | | | | | | ResNet56 | | | | | |
|---|---|---|---|---|---|---|---|---|---|---|---|---|
| Severity | S1 | | S3 | | S5 | | S1 | | S3 | | S5 | |
| Corruption | Before | After | Before | After | Before | After | Before | After | Before | After | Before | After |
| Gaussian | 33.18 | **59.17** | 11.76 | **32.13** | 8.26 | **19.78** | 37.06 | **64.47** | 13.13 | **34.46** | 8.11 | **22.97** |
| Shot | 43.97 | **62.23** | 15.93 | **40.91** | 9.09 | **21.84** | 49.32 | **67.11** | 18.18 | **43.72** | 9.62 | **25.31** |
| Impulse | 46.87 | **56.13** | 17.98 | **29.87** | 4.82 | **6.23** | 49.50 | **59.22** | 18.19 | **31.15** | 4.68 | **8.65** |
| Snow | 57.44 | **58.88** | 45.63 | **49.45** | 36.92 | **43.83** | 62.21 | **64.73** | 50.57 | **55.53** | 40.96 | **48.66** |
| Frost | 56.38 | **61.29** | 36.16 | **47.24** | 25.51 | **38.09** | 60.06 | **65.88** | 39.75 | **51.60** | 29.25 | **42.63** |
| Pixelate | 63.08 | 62.55 | 43.76 | **57.69** | 13.65 | **36.46** | 66.41 | **68.02** | 50.11 | **64.25** | 18.41 | **47.27** |
| JPEG | 51.48 | **60.49** | 40.74 | **56.31** | 33.90 | **52.65** | 53.36 | **64.43** | 42.89 | **60.25** | 35.21 | **57.05** |

| Backbone | RepVGG_a0 | | | | | | RepVGG_a2 | | | | | |
|---|---|---|---|---|---|---|---|---|---|---|---|---|
| Severity | S1 | | S3 | | S5 | | S1 | | S3 | | S5 | |
| Corruption | Before | After | Before | After | Before | After | Before | After | Before | After | Before | After |
| Gaussian | 38.56 | **68.31** | 11.62 | **42.52** | 6.99 | **28.42** | 39.44 | **68.46** | 12.76 | **40.54** | 7.83 | **27.61** |
| Shot | 51.35 | **70.67** | 17.45 | **50.86** | 8.30 | **30.92** | 54.18 | **71.05** | 19.19 | **49.61** | 9.49 | **29.28** |
| Impulse | 54.65 | **65.46** | 21.74 | **43.46** | 6.50 | **15.22** | 55.89 | **65.27** | 22.22 | **41.20** | 7.13 | **13.78** |
| Snow | 66.86 | **68.38** | 55.98 | **58.51** | 46.93 | **52.52** | 68.22 | **69.12** | 57.84 | **59.77** | 47.55 | **53.11** |
| Frost | 65.95 | **69.53** | 46.14 | **56.63** | 35.01 | **48.31** | 67.30 | **70.41** | 47.22 | **57.97** | 36.40 | **49.60** |
| Pixelate | 70.29 | **71.02** | 56.98 | **68.01** | 21.27 | **52.02** | 73.02 | **71.87** | 59.46 | **69.39** | 23.11 | **55.79** |
| JPEG | 60.58 | **68.09** | 50.55 | **64.76** | 43.81 | **61.38** | 62.55 | **69.28** | 52.31 | **65.89** | 44.88 | **62.98** |

having highly limited training data for each class, the proposed approach demonstrates improved performance across the network and maintains clean accuracy.

Table 6: Performance before and after applying the proposed method across common corruption types on the ImageNet subset using the AlexNet backbone.

| Method | Clean | Gaussian | Shot | Impulse | Snow | Frost | Pixelate | JPEG |
|---|---|---|---|---|---|---|---|---|
| Before | 91.0 | 53.89 | 52.30 | 49.46 | 63.77 | **70.57** | 73.41 | 85.21 |
| After | 91.01 | **56.29** | **54.54** | **51.37** | **67.26** | 67.18 | **80.16** | **88.03** |

## A.3 IMPLEMENTATION DETAILS

Our proposed framework, Dem-HEC, is implemented in PyTorch. We finetune all pretrained models for 20 epochs using a batch size of 128 and the full training sets of CIFAR10, CIFAR100, and Tiny-ImageNet. The training is performed using a Stochastic Gradient Descent (SGD) optimizer with an initial learning rate of 0.05, momentum of 0.9, and weight decay of $5 \times 10^{-4}$. A cosine annealing learning rate schedule with a 2-epoch linear warmup phase is employed. The parameter $\lambda_{KD} = 0.5$, and $T = 2.0$ has been taken for the $L_{KD}$. $\lambda_C = 1.0$ and temperature $\tau = 0.2$ has been taken for $L_{InfoNCE}$. The parameters for the CIFAR10 and CIFAR100 experiments are identical, while the Tiny-ImageNet experiment uses a smaller batch size=32, fewer epochs=10, and a lower learning rate =0.0005 to accommodate the larger Vision Transformer (ViT-L) model and higher resolution images = $384 \times 384$. All experiments are conducted on a machine with a 104-Core 2.0GHz CPU and 251GB system memory with an NVIDIA 47GB NVIDIA RTX A6000 GPU.

