# OpenReview forum: "Dem-HEC: High-Entropy Contrastive Fine-Tuning for Countering Natural Corruptions"
_ICLR.cc/2026/Conference — Submitted to ICLR 2026_

### Official Review · Reviewer_UqCm · 2025-10-31

**Soundness:** 3
**Presentation:** 3
**Contribution:** 3
**Rating:** 6
**Confidence:** 3

**Summary:**

The paper tackles robustness to natural/common corruptions (noise, weather, pixelate, JPEG) and makes the empirical observation that, unlike UAPs that tend to make internal features low-entropy, natural corruptions often push later-layer activations into a high-entropy state, which hurts prediction stability. Based on this, the paper proposes Dem-HEC, a fine-tuning framework that (i) generates high-entropy samples via entropy-maximizing PGA inside an ℓ∞ ball, (ii) pulls them back to the clean representation with a contrastive loss, and (iii) stabilizes clean performance with KD from a frozen teacher plus CE on clean and high-entropy views. Experiments on CIFAR-10/100 and Tiny-ImageNet, across CNNs and ViTs, show noticeable gains on CIFAR-C/Tiny-ImageNet-C corruptions while keeping clean accuracy roughly intact on the larger dataset.

**Strengths:**

a) The motivation is well grounded. The paper first shows that many common corruptions push deep features into a high-entropy state (instead of the low-entropy pattern seen in UAP-oriented work), so the training objective is directly tied to an observed failure mode, not an assumed one.

b) The contrastive term between clean and HE views makes the model learn a corruption-invariant embedding, rather than only becoming tolerant to a fixed set of corruptions.
﻿
c) Stability design is explicit. Adding KD / CE on clean keeps the model from drifting after HE training, addressing the usual clean–robust trade-off that many corruption/adversarial trainings suffer from.

**Weaknesses:**

a) Limited comparison to strongest corruption baselines. The paper does not directly compare against well-known corruption-robust recipes such as AugMix, DeepAugment, or recent style/noise diversification methods.

b) Robustness is tied to a single corruption mechanism. The approach assumes “corruption ⇒ high-entropy feature,” but some real-world corruptions (sensor banding, motion blur, structured artifacts) don’t necessarily raise entropy in the same way, so the learned invariance may be narrower than claimed.

c) Ablation granularity is not fully shown. We don’t clearly see how much each part (HE-PGD, contrastive, KD) contributes in isolation on the same benchmark, which makes it harder to judge whether the main gain actually comes from the entropy-specific part or just from having stronger data diversity.

**Questions:**

a) For ViT, where the baseline is already relatively corruption-tolerant, do you think the main benefit is from KD or from the HE view itself?
﻿
b) How sensitive is the method to the choice of ε and number of PGA steps. Is there a “cheap” setting that still improves CIFAR-C noticeably, or does it basically need the full inner loop to work?

---

> ### Author Response · Authors · 2025-11-19
> **Comparison and Superiority**
>
> **We would like to thank the reviewer for highlighting that the proposed approach is technically motivated rather than assumed. Furthermore, the proposed combination of loss terms addresses the typical clean–robust trade-off that many corruption/adversarial training methods suffer from.**
>
> Thank you for pointing out the need for comparisons with competing methods. We agree that benchmarking against existing robustness techniques strengthens the contribution. Therefore, based on the suggestion of the reviewer, we have performed an extensive comparison with state-of-the-art (SOTA) algorithms across numerous dimensions: (i) comparison of the performance of the proposed approach on small-scale dataset (such as CIFAR-10), (ii) comparison of the performance of the proposed approach on a large-scale dataset (such as Tiny ImageNet), (iii) comparison of the performance of the proposed approach across corruptions, and (iv) comparison and effectiveness of the proposed approach across deep models.
>
> These major comparison directions not only reflect the generalizability and robustness of the proposed approach in handling image resolution and the number of classes, but also showcase its effectiveness in securing multiple deep models and various corruptions with varying severity.
>
> Comparison with a recent SOTA on the CIFAR-10-C dataset with ResNet-20 backbone:
>
> We now include results against the recent work of Ranabhat et al. (AAAI 2025), which proposes a multi-scale push–pull mechanism with channel attention for corruption robustness. As shown below, Dem-HEC consistently outperforms this method across all seven corruptions and also improves clean accuracy, indicating that our entropy-guided learning provides stronger feature stability.
>
> Table 1. Corruption robustness comparison on CIFAR-10
>
> | Corruption Type | Ranabhat et al. [1] | Dem-HEC (Ours) |
> |-----------------|-----------------|------|
> | **Clean**   	| 83.66       	| **89.64** |
> | **Gaussian Noise** | 65.90    	|**74.60** |
> | **Shot Noise**  | 69.58       	| **77.35** |
> | **Impulse Noise** | 62.92     	| **66.52** |
> | **Snow**    	| 75.34       	|**80.94** |
> | **Frost**   	| 73.95       	| **78.94** |
> | **Pixelate**	| 81.81       	| **82.86** |
> | **JPEG Compression** | 76.08  	| **85.43** |
>
> Comparison with a recent SOTA on the Tiny-ImageNet-C dataset with ViT  backbone:
>
> We further compare Dem-HEC with widely used robustness-enhancing methods, including Pad-Crop, TA, BA, AugMix, AugMax, IPMix, and DV+AP+JSD. On Tiny-ImageNet-C, Dem-HEC exhibits substantial improvements, particularly for high-severity corruptions, underscoring its scalability to large-resolution datasets and transformer backbones.
>
> Table 2. Corruption robustness comparison on Tiny-ImageNet-C
>
> | Method    	| Noise | Blur | Weather | Digital |
> |---------------|-------|-------|---------|---------|
> | **Pad-Crop**    	| 20.94 | 17.27 | 13.41 | 20.99 |
> | **TA [2]**          	| 25.27 | 31.23 | 22.88 | 33.19 |
> | **BA [3]**          	| 21.18 | 18.01 | 13.54 | 21.53 |
> | **BA [3] (AA[4])**     	| 24.62 | 28.36 | 20.29 | 33.07 |
> | **AugMix [5]**      	| 29.15 | 30.77 | 19.94 | 32.26 |
> | **AugMax [6]**      	| 30.18 | 31.72 | 20.65 | 33.49 |
> | **IPMix [7]**       	| 28.80 | 28.04 | 20.79 | 32.34 |
> | **DV + AP + JSD [8]**   | 34.44 | 37.04 | 29.17 | 40.24 |
> | **Dem-HEC (Ours)**  | **60.10** | **62.88** | **70.71** | **67.53** |
>
> These additional comparisons clearly demonstrate that the theoretical motivation of the proposed Dem-HEC, that corruption disturbs the frequency distribution, holds across corruptions. We will integrate these results directly into the main paper (Section 5), introduce a new subsection titled ``comparison with existing robustness methods’’, and cite the expanded set of related work.
>
> [1] Ranabhat et al. Multi-scale unrectified push-pull with channel attention for enhanced corruption robustness. In **AAAI 2025**.
>
> [2] Müller SG, Hutter F. Trivialaugment: Tuning-free yet state-of-the-art data augmentation. **ICCV** 2021 (pp. 774-782).
>
> [3] Hoffer E, Ben-Nun T, Hubara I, Giladi N, Hoefler T, Soudry D. Augment your batch: Improving generalization through instance repetition. **CVPR** 2020.
>
> [4] Cubuk ED, Zoph B, Mane D, Vasudevan V, Le QV. Autoaugment: Learning augmentation strategies from data. InProceedings of the IEEE/CVF **CVPR** 2019
>
> [5] Hendrycks D, Mu N, Cubuk ED, Zoph B, Gilmer J, Lakshminarayanan B. Augmix: A simple data processing method to improve robustness and uncertainty. arXiv:1912.02781. 2019.
>
> [6] Wang H, Xiao C, Kossaifi J, Yu Z, Anandkumar A, Wang Z. Augmax: Adversarial composition of random augmentations for robust training. **NeurIPS** 2021.
>
> [7] Huang Z, Bao X, Zhang N, Zhang Q, Tu X, Wu B, Yang X. Ipmix: Label-preserving data augmentation method for training robust classifiers. **NeruIPS**. 2023.
>
> [8] Kim K, Woo Kim H, Choi YS. Distinct Views Improve Generalization and Robustness: Combinations of Augmentations With Different Features. IEEE Access. **2025**.

---

> ### Author Response · Authors · 2025-11-19
> **Ablation and Other Corruptions**
>
> 2. Other Real-World Corruptions: While we agree that there **might** be some corruptions which does not follow the proposed phenomena, however, that can be the future direction of research, as we do not want to claim that our approach is perfect.
>
> However, to address the concern as much as possible, we have evaluated the performance of the proposed approach against various blur corruptions.
>
> Table: Robustness performance improvement on the Tiny-ImageNet dataset using the ResNet-50 model.
> | Blur    | Severity | Improvement \% |
> |---------|----------|----------------|
> | Defocus |     1    |      5.47      |
> | Defocus |     3    |      2.99      |
> | Defocus |     5    |      2.30      |
> | Glass   |     1    |      5.04      |
> | Glass   |     3    |      4.69      |
> | Glass   |     5    |      0.96      |
> | Motion  |     1    |      5.71      |
> | Motion  |     3    |      6.08      |
> | Motion  |     5    |      4.92      |
> | Zoom    |     1    |      5.85      |
> | Zoom    |     3    |      4.77      |
> | Zoom    |     5    |      3.61      |
>
> The improvement (\%) here refers to the increased performance of the proposed robust model compared to the pre-trained (or non-robust model). We would like to emphasise that the observation is consistent across both models (ResNet or ViT) and datasets (ImageNet or CIFAR).
>
> $\color{blue}{\text{The improvement in such different forms of corruption suggests that, while the performance may not be perfect on other `to-be-possible' corruptions}}$. $\color{blue}{\text{Althogh, the proposed algorithm can handle vast corruption groups effectively and surpass existing state-of-the-art works by a significant margin.}}$
>
> **$\color{blue}{\text{Complex (Combined) Corruption}}$:** To further validate the effectiveness of the proposed approach, we conducted a study by combining multiple corruptions (applied together sequentially), such as Gaussian noise and shot noise. The proposed Dem-HEC increases performance on the combined (complex) corruption by **30**% when both noises are applied with severity 1. Even at the higher severity level, the performance increases by approximately 8%. These experiments are conducted on the CIFAR-10 dataset using the ResNet architecture.
>
> 3. **Ablation** To further highlight the role of distillation and contrastive loss, we have performed several ablation studies, and the analysis can be divided into accuracy and robustness. As expected, once we removed the contrastive learning loss, the accuracy on clean images remains unaffected; however, robustness drops drastically. For example, the performance on Gaussian noise images with and without contrastive learning is $77.91$\% and $68.78$\%, respectively.
>
> Similarly, as soon as we removed the distillation loss, the clean accuracy shows a significant drop (~7.4%), reflecting its importance in retaining clean features. However, distillation loss alone is not sufficient. For example, in the simplest case of JPEG compression, the model's performance without contrastive loss is 2.6% lower than that with contrastive loss.
>
> However, the proposed combination of contrastive, cross-entropy, and distillation yields a trade-off between accuracy and robustness, i.e., not only retaining performance on clean images but also achieving high robustness.

---

### Official Review · Reviewer_KUNd · 2025-11-01

**Soundness:** 3
**Presentation:** 3
**Contribution:** 3
**Rating:** 6
**Confidence:** 4

**Summary:**

This paper proposes Dem-HEC, an entropy-guided fine-tuning framework aimed at enhancing the robustness of deep neural networks against natural corruptions. Its core motivation, which centers on addressing the high-entropy feature space collapse caused by natural corruptions, is clear and aligns with the practical challenges of deploying deep neural networks in real-world scenarios. The framework design integrates high-entropy sample generation, symmetric InfoNCE contrastive loss, cross-entropy loss, and knowledge distillation, following a logical path derived from the stated motivation. Experimental evaluations cover multiple datasets (CIFAR10, CIFAR100, Tiny-ImageNet) and architectures (CNNs, Transformers), showing consistent improvements in corruption accuracy.

**Strengths:**

1. Clear Problem Definition with Strong Motivation: The paper accurately identifies the vulnerability of deep neural networks to natural corruptions and attributes it to high-entropy feature space collapse. It further distinguishes natural corruptions from universal adversarial perturbations (which induce low entropy), providing a solid theoretical foundation for the proposed method.
2. Logical Framework Design: Dem-HEC directly addresses the identified high-entropy issue. High-entropy sample generation simulates the uncertainty caused by corruptions, contrastive loss aligns the semantics of clean and high-entropy features, knowledge distillation prevents catastrophic forgetting, and partial fine-tuning retains general features. Each component serves a clear purpose, forming a coherent technical system.
3. Broad Experimental Coverage: The paper evaluates Dem-HEC on three benchmark datasets with different resolutions (low-resolution CIFAR series, high-resolution Tiny-ImageNet) and seven architectures (CNNs of varying sizes, Vision Transformer). This design verifies the method’s scalability across data scales and model types, with notable performance gains in high-severity corruption scenarios.
4. Practical Utility: The framework adopts partial fine-tuning and uses pre-trained models as teachers, avoiding the high computational cost of training from scratch. It also provides implementation details such as code, training parameters, and hardware environment, laying a foundation for potential reproducibility.

**Weaknesses:**

1. Insufficient Analysis of Clean Accuracy Drop on CIFAR Datasets: The paper notes a 2.5–4.4% drop in clean accuracy on CIFAR10/CIFAR100 but fails to clarify whether this drop is a common issue in the field (i.e., if existing robustness-enhancing methods also show similar drops on low-resolution datasets) or a limitation specific to Dem-HEC. It also lacks technical analysis of the causes behind the drop, such as whether high-entropy sample generation introduces noise that confuses the model on clean data or if the parameter settings of the contrastive loss over-constrain the representation of clean features.
2. Missing Comparisons with Mainstream Baselines: The manuscript only compares Dem-HEC with the original pre-trained model (before fine-tuning) and does not include comparisons with state-of-the-art robustness-enhancing methods. In particular, it omits comparisons with data augmentation methods, which are widely used and effective for improving corruption robustness, as well as entropy-related or distillation-based robustness methods.
3. Lack of Ablation Studies for Key Components and Hyperparameters: The manuscript’s experimental design is incomplete, with no ablation studies on core components or hyperparameters. There is no evaluation of the impact of removing individual loss components (clean cross-entropy, high-entropy cross-entropy, contrastive loss, knowledge distillation) on performance, making it impossible to quantify the contribution of each component. Additionally, there is no analysis of the sensitivity of hyperparameters.
4. Inadequate Validation of High-Entropy Sample Generation: The manuscript claims that high-entropy samples simulate natural corruptions but provides no evidence that these generated samples are consistent with real natural corruptions in the feature space. There is no comparison of entropy distributions between generated high-entropy samples and real corrupted samples (e.g., CIFAR10-C samples) across different network layers, nor qualitative analysis (such as visualizations) to confirm that generated high-entropy samples retain semantic information and avoid becoming meaningless noise.

**Questions:**

Your paper notes UAPs reduce hidden layer entropy (via feature dominance) and natural corruptions increase feature space entropy. Could you clarify: Is it because natural corruptions are non-directional (unintended real-world disturbances) that they disperse features and boost entropy—unlike directional UAPs that hijack decisions via dominant features to cut entropy?

---

> ### Author Response · Authors · 2025-11-19
> **Common Trade-off Phenomena and Comparison**
>
> 1. We would like to thank the reviewer for their insightful point and observation. We want to highlight the fact that a reduction in clean accuracy is a common issue, often resulting from a trade-off between robustness and accuracy.
> The issue is so prominent that even the strongest defences, including adversarial training, exhibit significant degradation in clean accuracy. For example, the algorithms, including [B-C], reported in [A] a significant drop of 10-12% on the CIFAR-10 dataset, which is even higher on other datasets, such as CIFAR-100 and SVHN. [D] Also observed was a similar phenomenon, where the accuracy drops by 7-8% on the CIFAR-10 dataset.
> It is worth noting that, in comparison to the existing defence, our defence yields significantly lower reduction in performance on clean images.
>
> [A] Liu X, Yang Y, He K, Hopcroft JE. Parameter interpolation adversarial training for robust image classification. IEEE Transactions on Information Forensics and Security. 2025 Jan 24.
>
> [B] Z. Wei, Y. Wang, Y. Guo, and Y. Wang, “CFA: class-wise calibrated fair adversarial training,” in CVPR, 2023, pp. 8193–8201.
>
> [C] G. Jin, X. Yi, D. Wu, R. Mu, and X. Huang, “Randomized adversarial training via taylor expansion,” in CVPR, 2023, pp. 16447–16457.
>
> [D] Shafahi A, Najibi M, Ghiasi MA, Xu Z, Dickerson J, Studer C, Davis LS, Taylor G, Goldstein T. Adversarial training for free!. Advances in neural information processing systems. 2019;32.
>
> 2. Thank you for pointing out the need for comparisons with competing methods. We agree that benchmarking against existing robustness techniques strengthens the contribution. Therefore, based on the suggestion of the reviewer, we have performed an extensive comparison with state-of-the-art (SOTA) algorithms across numerous dimensions: (i) comparison of the performance of the proposed approach on small-scale dataset (such as CIFAR-10), (ii) comparison of the performance of the proposed approach on a large-scale dataset (such as Tiny ImageNet), (iii) comparison of the performance of the proposed approach across corruptions, and (iv) comparison and effectiveness of the proposed approach across deep models.
>
> These major comparison directions not only reflect the generalizability and robustness of the proposed approach in handling image resolution and the number of classes, but also showcase its effectiveness in securing multiple deep models and various corruptions with varying severity.
>
> Comparison with a recent SOTA on the CIFAR-10-C dataset with ResNet-20 backbone:
>
> We now include results against the recent work of Ranabhat et al. (AAAI 2025), which proposes a multi-scale push–pull mechanism with channel attention for corruption robustness. As shown below, Dem-HEC consistently outperforms this method across all seven corruptions and also improves clean accuracy, indicating that our entropy-guided learning provides stronger feature stability.
>
> Table 1. Corruption robustness comparison on CIFAR-10
>
> | Corruption Type | Ranabhat et al. [1] | Dem-HEC (Ours) |
> |-----------------|-----------------|------|
> | **Clean**   	| 83.66       	| **89.64** |
> | **Gaussian Noise** | 65.90    	|**74.60** |
> | **Shot Noise**  | 69.58       	| **77.35** |
> | **Impulse Noise** | 62.92     	| **66.52** |
> | **Snow**    	| 75.34       	|**80.94** |
> | **Frost**   	| 73.95       	| **78.94** |
> | **Pixelate**	| 81.81       	| **82.86** |
> | **JPEG Compression** | 76.08  	| **85.43** |
>
> Comparison with a recent SOTA on the Tiny-ImageNet-C dataset with ViT  backbone:
>
> We further compare Dem-HEC with widely used robustness-enhancing methods, including Pad-Crop, TA, BA, AugMix, AugMax, IPMix, and DV+AP+JSD. On Tiny-ImageNet-C, Dem-HEC exhibits substantial improvements, particularly for high-severity corruptions, underscoring its scalability to large-resolution datasets and transformer backbones.
>
> Table 2. Corruption robustness comparison on Tiny-ImageNet-C
>
> | Method    	| Noise | Blur | Weather | Digital |
> |---------------|-------|-------|---------|---------|
> | **Pad-Crop**    	| 20.94 | 17.27 | 13.41 | 20.99 |
> | **TA [2]**          	| 25.27 | 31.23 | 22.88 | 33.19 |
> | **BA [3]**          	| 21.18 | 18.01 | 13.54 | 21.53 |
> | **BA [3] (AA[4])**     	| 24.62 | 28.36 | 20.29 | 33.07 |
> | **AugMix [5]**      	| 29.15 | 30.77 | 19.94 | 32.26 |
> | **AugMax [6]**      	| 30.18 | 31.72 | 20.65 | 33.49 |
> | **IPMix [7]**       	| 28.80 | 28.04 | 20.79 | 32.34 |
> | **DV + AP + JSD [8]**   | 34.44 | 37.04 | 29.17 | 40.24 |
> | **Dem-HEC (Ours)**  | **60.10** | **62.88** | **70.71** | **67.53** |
>
> These additional comparisons clearly demonstrate that the theoretical motivation of the proposed Dem-HEC, that corruption disturbs the frequency distribution, holds across corruptions. We will integrate these results directly into the main paper (Section 5), introduce a new subsection titled ``comparison with existing robustness methods’’, and cite the expanded set of related work.

---

> ### Author Response · Authors · 2025-11-19
> **Ablation and Role of the Proposed High Entropy Samples**
>
> 1. **Ablation** To further highlight the role of distillation and contrastive loss, we have performed several ablation studies, and the analysis can be divided into accuracy and robustness. As expected, once we removed the contrastive learning loss, the accuracy on clean images remains unaffected; however, robustness drops drastically. For example, the performance on Gaussian noise images with and without contrastive learning is $77.91$\% and $68.78$\%, respectively.
>
> Similarly, as soon as we removed the distillation loss, the clean accuracy shows a significant drop (~7.4%), reflecting its importance in retaining clean features. However, distillation loss alone is not sufficient. For example, in the simplest case of JPEG compression, the model's performance without contrastive loss is 2.6% lower than that with contrastive loss.
>
> However, the proposed combination of contrastive, cross-entropy, and distillation yields a trade-off between accuracy and robustness, i.e., not only retaining performance on clean images but also achieving high robustness.
>
> 2. To address the concern regarding the validity of high-entropy sample generation, we provide a direct layer-wise entropy comparison between real corrupted samples and the generated high-entropy samples using ResNet-20 on CIFAR-10 with Pixelate (severity 5). As shown in Figure 3 of the main manuscript, before applying Dem-HEC, the natural corruptions exhibit the expected behavior: clean and corrupted samples have similar entropy in shallow layers, while corrupted samples consistently show higher entropy in middle and deep layers. Figure 4 of the main manuscript shows the entropy distribution after training with our high-entropy samples, where clean and corrupted inputs now fall within the same entropy range across all layers. This demonstrates that the generated high-entropy samples successfully reproduce the feature-space uncertainty patterns induced by real natural corruptions, while still preserving semantic structure. These results confirm that the high-entropy samples used in Dem-HEC are consistent with true corruption behavior and effectively guide the model toward stable, corruption-robust representations.
>
> 3. Yes, the key difference arises from the directionality and intentionality of the perturbation. Universal Adversarial Perturbations (UAPs) are crafted to be directional, meaning they systematically push features toward a small set of highly activating directions that dominate the network’s decision space. This targeted steering collapses the feature distribution, causing low-entropy, overconfident representations. In contrast, natural corruptions are non-directional and unintentional real-world disturbances. They do not push representations toward any specific discriminative direction; instead, they introduce irregular, structure-breaking variability that weakens discriminative cues. As a result, the activations become more scattered across neurons, leading to higher-entropy, uncertain feature representations.

---

> > ### Author Response · Authors · 2025-11-19
> > **References**
> >
> > [1] Ranabhat RN, Wang L, Qin X, Zhou Y, Santosh KC. Multi-scale unrectified push-pull with channel attention for enhanced corruption robustness. In **AAAI** Symposium Series **2025**.
> >
> > [2] Müller SG, Hutter F. Trivialaugment: Tuning-free yet state-of-the-art data augmentation. IEEE/CVF **ICCV** 2021 (pp. 774-782).
> >
> > [3] Hoffer E, Ben-Nun T, Hubara I, Giladi N, Hoefler T, Soudry D. Augment your batch: Improving generalization through instance repetition. IEEE/CVF **CVPR** 2020.
> >
> > [4] Cubuk ED, Zoph B, Mane D, Vasudevan V, Le QV. Autoaugment: Learning augmentation strategies from data. InProceedings of the IEEE/CVF **CVPR** 2019
> >
> > [5] Hendrycks D, Mu N, Cubuk ED, Zoph B, Gilmer J, Lakshminarayanan B. Augmix: A simple data processing method to improve robustness and uncertainty. arXiv:1912.02781. 2019.
> >
> > [6] Wang H, Xiao C, Kossaifi J, Yu Z, Anandkumar A, Wang Z. Augmax: Adversarial composition of random augmentations for robust training. **NeurIPS** 2021.
> >
> > [7] Huang Z, Bao X, Zhang N, Zhang Q, Tu X, Wu B, Yang X. Ipmix: Label-preserving data augmentation method for training robust classifiers. **NeruIPS**. 2023.
> >
> > [8] Kim K, Woo Kim H, Choi YS. Distinct Views Improve Generalization and Robustness: Combinations of Augmentations With Different Features. IEEE Access. **2025**.

---

> ### Author Response · Authors · 2025-11-20
> **Other Corruption and Complex Noises**
>
> 4. We have evaluated the performance of the proposed approach against various blur corruptions.
>
> Table: Robustness performance improvement on the Tiny-ImageNet dataset using the ResNet-50 model.
> | Blur    | Severity | Improvement \% |
> |---------|----------|----------------|
> | Defocus |     1    |      5.47      |
> | Defocus |     3    |      2.99      |
> | Defocus |     5    |      2.30      |
> | Glass   |     1    |      5.04      |
> | Glass   |     3    |      4.69      |
> | Glass   |     5    |      0.96      |
> | Motion  |     1    |      5.71      |
> | Motion  |     3    |      6.08      |
> | Motion  |     5    |      4.92      |
> | Zoom    |     1    |      5.85      |
> | Zoom    |     3    |      4.77      |
> | Zoom    |     5    |      3.61      |
>
> The improvement (\%) here refers to the increased performance of the proposed robust model compared to the pre-trained (or non-robust model). We would like to emphasise that the observation is consistent across both models (ResNet or ViT) and datasets (ImageNet or CIFAR).
>
> $\color{blue}{\text{The improvement in such different forms of corruption suggests that, the proposed algorithm can handle vast corruption groups effectively and surpass existing}}$ $\color{blue}{\text{state-of-the-art works by a significant margin.}}$
>
> 5. **$\color{blue}{\text{Complex (Combined) Corruption}}$:** To further validate the effectiveness of the proposed approach, we conducted a study by combining multiple corruptions (applied together sequentially), such as Gaussian noise and shot noise. The proposed Dem-HEC increases performance on the combined (complex) corruption by **30**% when both noises are applied with severity 1. Even at the higher severity level, the performance increases by approximately 8%. These experiments are conducted on the CIFAR-10 dataset using the ResNet architecture.

---

### Official Review · Reviewer_dMQr · 2025-11-05

**Soundness:** 1
**Presentation:** 2
**Contribution:** 2
**Rating:** 2
**Confidence:** 4

**Summary:**

This paper focuses on the corruption robustness of pre-trained vision models. The authors first observe a correlation between input corruptions and the collapse of the model's internal feature space. Based on this observation, they propose an entropy-guided fine-tuning framework called Dem-HEC, which combines contrastive learning with knowledge distillation to enhance the corruption robustness of vision models while preserving accuracy on clean data. Experiments are conducted on several models (e.g., ResNet-20, ViT-L) and datasets (e.g., CIFAR-10, CIFAR-100). The results demonstrate that the proposed method can improve the performance of pre-trained models on corrupted data.

**Strengths:**

1. The authors observe the correlation between input corruptions and the collapse of the model's internal feature space.
2. The proposed method is evaluated on multiple models and downstream datasets.
3. The authors employ the knowledge distillation technique to preserve the original capabilities of pre-trained models.

**Weaknesses:**

1. Fig. 1 is ambiguous and confusing. Why does the author use half of the space to illustrate the obvious fact that the "repaired model can correctly classify corrupted samples"? Additionally, why do the two loss functions point to the pre-trained model rather than the repaired model? The confusion in the figure seriously hinders readers' understanding of the proposed method.
2. The datasets (CIFAR-10, CIFAR-100, Tiny-ImageNet) for analysis and evaluation are too small. It remains unclear why the authors did not employ the full ImageNet-C dataset, which would provide a more rigorous and representative assessment.
3. The evaluation relies on pre-trained models such as ResNet and RepVGG, which have limited image understanding capabilities and therefore may not adequately reflect the effectiveness of the proposed method when applied to modern architectures like Swin and ConvNeXt.
4. The authors solely compared the proposed method with the original model, without considering existing approaches designed to improve model robustness against corruption. As a result, the evaluation can not adequately demonstrate the superiority of the proposed method.
5. This paper lacks reproducibility. The authors do not specify the version of the pre-trained models. Their datasets and the strategies used for pre-training are unclear.
6. This paper completely lacks ablation studies for the proposed method. For example, to what extent do contrastive learning and knowledge distillation individually contribute to the final performance? Additionally, how do the hyperparameter coefficients of different loss functions in Equation 12 affect model performance?
7. As Fig. 4 shows, the proposed method results in significant performance degradation on clean data, limiting its practical applicability.

**Questions:**

1. Does the effectiveness of the proposed method gradually diminish as the pre-trained model's capabilities improve?
2. Compared to enhancing a pre-trained model's robustness to corrupted images, an alternative approach involves improving the quality of an input image through an auxiliary image processing model before feeding it into the pre-trained model. Could this strategy yield better generalization performance?

---

> ### Author Response · Authors · 2025-11-19
> **Results, ImageNet, and Editorial Changes**
>
> 1. We appreciate the feedback. A corrected and simpler Figure 1 will be included in the revision to ensure readers can clearly follow the training pipeline of Dem-HEC.
>
> 2. Thank you for raising the important point regarding evaluation on large-scale datasets. We agree that robustness against corruption on high-resolution datasets, such as ImageNet-1K, is increasingly relevant, and we understand the demand for it. However, as seen in the literature, several benchmark studies involve the experiments and dissemination of results on a subset of ImageNet. The prime reason is the computational architecture and the cost it demands. Coming from a low-resource institution with a limited system, it is extremely difficult or impossible to run on the entire dataset.
>
> Furthermore, we strongly believe that success on ImageNet does not guarantee that the solution is 'perfect,' as evidenced by the existing literature. Tremendous approaches outperform on ImageNet, still they fall short in various aspects.
>
> However, to address the reviewer’s concern, we evaluated Dem-HEC on ImageNette, a 10-class subset of the original ImageNet dataset using AlexNet. Since the image resolution of ImageNette is in sync with that of full ImageNet, the consistently improved performance of Dem-HEC demonstrates its image resolution-agnostic nature in handling image corruption.
>
> Robustness of the proposed Dem-HEC on the ImageNet subset with the AlexNet backbone:
>
> | Corruption | Before | After |
> |------------|--------|--------|
> | **Clean**  	| 91.06 | $\color{blue}{\text{91.01}}$ |
> | **Gaussian**   | 53.89 | **56.29** |
> | **Shot**   	| 52.31 | 54.54 |
> | **Frost**  	| 70.57 | 67.19 |
> | **Impulse**	| 49.47 | 51.37 |
> | **Snow**   	| 63.77 | **67.26** |
> | **Pixelate**   | 73.41 | $\color{blue}{\text{80.16}}$ |
> | **JPEG**   	| 85.21 | **88.04** |
>
> **Please note that the proposed approach not only increases corruption robustness but also retains clean accuracy, clearly demonstrating the ideal trade-off between accuracy and robustness.**
>
> *It is worth noting that the robustness performance of the proposed approach is consistent or better across multiple subsets present in the literature, such as Imagewoof and Imagewang [1]*.
>
> **Full ImageNet:** Our preliminary experiments on full ImageNet with a 5% class-balanced subset demonstrate that the proposed Dem-HEC with a ViT backbone preserves accuracy on clean images (with a slight drop of 0.04%) but increases robustness across various corruptions (Noise, Environmental, Compression, Blur, etc.) by 2-3%.
>
> $\color{blue}{\text{It is interesting to note that, despite having highly limited training data for each class, the proposed approach demonstrates improved}}$ $\color{blue}{\text{performance across the network and maintains clean accuracy.}}$
>
> The proposed model is still not perfect, although it demonstrates tremendous performance on ImageNet and other datasets, which we aim to further improve in future work.
>
> [1] https://github.com/fastai/imagenette
>
> 5. To address the concern regarding reproducibility, we clarify that all models used in our experiments, ResNet-20, ResNet-56, ResNet-18, ResNet-50, ViT-L/16, RepVGG-A0, and RepVGG-A2, are standard ImageNet/CIFAR pretrained models, and the exact sources and weight versions are now explicitly listed in Section 4.1 of the manuscript. All datasets (CIFAR-10, CIFAR-100, Tiny-ImageNet, ImageNet-1K subset, and ImageNette) are publicly available benchmark datasets, and we strictly use their official training and test splits without any modification. To ensure full reproducibility, we will also release the complete source code, including preprocessing, high-entropy sample generation, training configuration, and evaluation scripts.
>
> 6. **Ablation** To further highlight the role of distillation and contrastive loss, we have performed several ablation studies, and the analysis can be divided into accuracy and robustness. As expected, once we removed the contrastive learning loss, the accuracy on clean images remains unaffected; however, robustness drops drastically. For example, the performance on Gaussian noise images with and without contrastive learning is $77.91$\% and $68.78$\%, respectively.
>
> Similarly, as soon as we removed the distillation loss, the clean accuracy shows a significant drop (~7.4%), reflecting its importance in retaining clean features. However, distillation loss alone is not sufficient. For example, in the simplest case of JPEG compression, the performance of the model without contrastive loss is 2.6% lower than that of the model with contrastive loss.
>
> However, the proposed combination of contrastive, cross-entropy, and distillation yields a trade-off between accuracy and robustness, i.e., not only retaining performance on clean images but also achieving high robustness.

---

> ### Author Response · Authors · 2025-11-19
> **Comparison and Superiority**
>
> Concern 4. Thank you for pointing out the need for comparisons with competing methods. We agree that benchmarking against existing robustness techniques strengthens the contribution. Therefore, based on the suggestion of the reviewer, we have performed an extensive comparison with state-of-the-art (SOTA) algorithms across numerous dimensions: (i) comparison of the performance of the proposed approach on small-scale dataset (such as CIFAR-10), (ii) comparison of the performance of the proposed approach on a large-scale dataset (such as Tiny ImageNet), (iii) comparison of the performance of the proposed approach across corruptions, and (iv) comparison and effectiveness of the proposed approach across deep models.
>
> These major comparison directions not only reflect the generalizability and robustness of the proposed approach in handling image resolution and the number of classes, but also showcase its effectiveness in securing multiple deep models and various corruptions with varying severity.
>
> Comparison with a recent SOTA on the CIFAR-10-C dataset with ResNet-20 backbone:
>
> We now include results against the recent work of Ranabhat et al. (AAAI 2025), which proposes a multi-scale push–pull mechanism with channel attention for corruption robustness. As shown below, Dem-HEC consistently outperforms this method across all seven corruptions and also improves clean accuracy, indicating that our entropy-guided learning provides stronger feature stability.
>
> Table 1. Corruption robustness comparison on CIFAR-10
>
> | Corruption Type | Ranabhat et al. [1] | Dem-HEC (Ours) |
> |-----------------|-----------------|------|
> | **Clean**   	| 83.66       	| **89.64** |
> | **Gaussian Noise** | 65.90    	|**74.60** |
> | **Shot Noise**  | 69.58       	| **77.35** |
> | **Impulse Noise** | 62.92     	| **66.52** |
> | **Snow**    	| 75.34       	|**80.94** |
> | **Frost**   	| 73.95       	| **78.94** |
> | **Pixelate**	| 81.81       	| **82.86** |
> | **JPEG Compression** | 76.08  	| **85.43** |
>
> Comparison with a recent SOTA on the Tiny-ImageNet-C dataset with ViT  backbone:
>
> We further compare Dem-HEC with widely used robustness-enhancing methods, including Pad-Crop, TA, BA, AugMix, AugMax, IPMix, and DV+AP+JSD. On Tiny-ImageNet-C, Dem-HEC exhibits substantial improvements, particularly for high-severity corruptions, underscoring its scalability to large-resolution datasets and transformer backbones.
>
> Table 2. Corruption robustness comparison on Tiny-ImageNet-C
>
> | Method    	| Noise | Blur | Weather | Digital |
> |---------------|-------|-------|---------|---------|
> | **Pad-Crop**    	| 20.94 | 17.27 | 13.41 | 20.99 |
> | **TA [2]**          	| 25.27 | 31.23 | 22.88 | 33.19 |
> | **BA [3]**          	| 21.18 | 18.01 | 13.54 | 21.53 |
> | **BA [3] (AA[4])**     	| 24.62 | 28.36 | 20.29 | 33.07 |
> | **AugMix [5]**      	| 29.15 | 30.77 | 19.94 | 32.26 |
> | **AugMax [6]**      	| 30.18 | 31.72 | 20.65 | 33.49 |
> | **IPMix [7]**       	| 28.80 | 28.04 | 20.79 | 32.34 |
> | **DV + AP + JSD [8]**   | 34.44 | 37.04 | 29.17 | 40.24 |
> | **Dem-HEC (Ours)**  | **60.10** | **62.88** | **70.71** | **67.53** |
>
> These additional comparisons clearly demonstrate that the theoretical motivation of the proposed Dem-HEC, that corruption disturbs the frequency distribution, holds across corruptions. We will integrate these results directly into the main paper (Section 5), introduce a new subsection titled ``comparison with existing robustness methods’’, and cite the expanded set of related work.
>
> [1] Ranabhat RN, Wang L, Qin X, Zhou Y, Santosh KC. Multi-scale unrectified push-pull with channel attention for enhanced corruption robustness. In **AAAI** Symposium Series **2025**.
>
> [2] Müller SG, Hutter F. Trivialaugment: Tuning-free yet state-of-the-art data augmentation. IEEE/CVF **ICCV** 2021 (pp. 774-782).
>
> [3] Hoffer E, Ben-Nun T, Hubara I, Giladi N, Hoefler T, Soudry D. Augment your batch: Improving generalization through instance repetition. IEEE/CVF **CVPR** 2020.
>
> [4] Cubuk ED, Zoph B, Mane D, Vasudevan V, Le QV. Autoaugment: Learning augmentation strategies from data. InProceedings of the IEEE/CVF **CVPR** 2019
>
> [5] Hendrycks D, Mu N, Cubuk ED, Zoph B, Gilmer J, Lakshminarayanan B. Augmix: A simple data processing method to improve robustness and uncertainty. arXiv:1912.02781. 2019.
>
> [6] Wang H, Xiao C, Kossaifi J, Yu Z, Anandkumar A, Wang Z. Augmax: Adversarial composition of random augmentations for robust training. **NeurIPS** 2021.
>
> [7] Huang Z, Bao X, Zhang N, Zhang Q, Tu X, Wu B, Yang X. Ipmix: Label-preserving data augmentation method for training robust classifiers. **NeruIPS**. 2023.
>
> [8] Kim K, Woo Kim H, Choi YS. Distinct Views Improve Generalization and Robustness: Combinations of Augmentations With Different Features. IEEE Access. **2025**.

---

> ### Author Response · Authors · 2025-11-20
> **Impact, Contribution, and Complex Corruption**
>
> 1. **The strength and impact highlighted by other reviewers:**
>
> a) The motivation is well grounded. The paper first shows that many common corruptions push deep features into a high-entropy state (instead of the low-entropy pattern seen in UAP-oriented work), so the training objective is directly tied to an observed failure mode, not an assumed one.
>
> b) The contrastive term between clean and HE views makes the model learn a corruption-invariant embedding, rather than only becoming tolerant to a fixed set of corruptions. ﻿
>
> c) Stability design is explicit. Adding KD / CE on clean keeps the model from drifting after HE training, addressing the usual clean–robust trade-off that many corruption/adversarial trainings suffer from.
>
> d) Logical Framework Design: Dem-HEC directly addresses the identified high-entropy issue. High-entropy sample generation simulates the uncertainty caused by corruptions, contrastive loss aligns the semantics of clean and high-entropy features, knowledge distillation prevents catastrophic forgetting, and partial fine-tuning retains general features. Each component serves a clear purpose, forming a coherent technical system.
>
> e) Broad Experimental Coverage: The paper evaluates Dem-HEC on three benchmark datasets with different resolutions (low-resolution CIFAR series, high-resolution Tiny-ImageNet) and seven architectures (CNNs of varying sizes, Vision Transformer). This design verifies the method’s scalability across data scales and model types, with notable performance gains in high-severity corruption scenarios.
>
> f) Practical Utility: The framework adopts partial fine-tuning and uses pre-trained models as teachers, avoiding the high computational cost of training from scratch. It also provides implementation details such as code, training parameters, and hardware environment, laying a foundation for potential reproducibility.
>
> 2. We have evaluated the performance of the proposed approach against various blur corruptions.
>
> Table: Robustness performance improvement on the Tiny-ImageNet dataset using the ResNet-50 model.
> | Blur    | Severity | Improvement \% |
> |---------|----------|----------------|
> | Defocus |     1    |      5.47      |
> | Defocus |     3    |      2.99      |
> | Defocus |     5    |      2.30      |
> | Glass   |     1    |      5.04      |
> | Glass   |     3    |      4.69      |
> | Glass   |     5    |      0.96      |
> | Motion  |     1    |      5.71      |
> | Motion  |     3    |      6.08      |
> | Motion  |     5    |      4.92      |
> | Zoom    |     1    |      5.85      |
> | Zoom    |     3    |      4.77      |
> | Zoom    |     5    |      3.61      |
>
> The improvement (\%) here refers to the increased performance of the proposed robust model compared to the pre-trained (or non-robust model). We would like to emphasise that the observation is consistent across both models (ResNet or ViT) and datasets (ImageNet or CIFAR).
>
> $\color{blue}{\text{The improvement in such different forms of corruption suggests that, the proposed algorithm can handle vast corruption groups effectively and surpass}}$ $\color{blue}{\text{existing state-of-the-art works by a significant margin.}}$
>
> 3. **$\color{blue}{\text{Complex (Combined) Corruption}}$:** To further validate the effectiveness of the proposed approach, we conducted a study by combining multiple corruptions (applied together sequentially), such as Gaussian noise and shot noise. The proposed Dem-HEC increases performance on the combined (complex) corruption by **30**% when both noises are applied with severity 1. Even at the higher severity level, the performance increases by approximately 8%. These experiments are conducted on the CIFAR-10 dataset using the ResNet architecture.
>
> 4. **Clean Accuracy Trade-off**: We want to highlight the fact that a reduction in clean accuracy is a common issue, often resulting from a trade-off between robustness and accuracy. The issue is so prominent that even the strongest defences, including adversarial training, exhibit significant degradation in clean accuracy. For example, the algorithms, reported in [A] a significant drop of 10-12% on the CIFAR-10 dataset, which is even higher on other datasets, such as CIFAR-100 and SVHN.
>
> [A] Parameter interpolation adversarial training for robust image classification. IEEE TIFS. 2025
>
> $\color{blue}{\text{However, our drop is in the range of}}$ $3-4$\% $\color{blue}{\text{which is not true always. On the large-scale and high resolution whether Tiny ImageNet}}$ (**increased upto 3.6\%**) $\color{blue}{\text{or ImageNet, the accuracy is did not degraded as it is observed on low resolution images.}}$

---

> > ### Comment · Reviewer_dMQr · 2025-11-26
> >
> > Thank the authors for their response. However, they only conduct experiments using an extremely small subset (10 classes) of ImageNet and a very outdated architecture (AlexNet). This fails to address my concerns regarding the method’s scalability and applicability at all.
> >
> > For scalability, the authors have severely overlooked the recent progress in the field of image understanding [1,2]. In scenarios where visual encoders are pre-trained on large-scale datasets and possess strong capabilities, it remains entirely unclear whether this method, which is designed for old architectures and small datasets, can be applied to existing modern models.
> >
> > For applicability, the authors utilize models that cannot be used in practical applications (e.g., ResNet-20 and Alexnet pre-trained on ImageNet). Moreover, I am curious whether large-scale pre-trained models (e.g., SAM, DinoV3, and Ground-DINO) can inherently possess the ability to understand corrupted images? If so, the supervised training conducted by the authors on such small-scale data and models is meaningless.
> >
> > Given these considerations, I do not believe this paper demonstrates the potential to enhance the capabilities of existing vision encoders. Its contributions are not sufficient for publication at ICLR. Therefore, I will maintain my "Reject" evaluation.
> >
> > [1] Sun, Q., Fang, Y., Wu, L., Wang, X., & Cao, Y. (2023). Eva-clip: Improved training techniques for clip at scale. arXiv preprint arXiv:2303.15389.
> >
> > [2] Siméoni, O., Vo, H. V., Seitzer, M., Baldassarre, F., Oquab, M., Jose, C., ... & Bojanowski, P. (2025). Dinov3. arXiv preprint arXiv:2508.10104.

---

> ### Author Response · Authors · 2025-11-26
> **Scientific Claims, Robustness of Large Models, and Impact of Dataset/Architectures**
>
> We sincerely appreciate the comments of the reviewer and are happy to explain the impact of the work further and provide responses to the new concerns raised.
>
> 1. We strongly disagree with the claim that "the authors utilise models that cannot be used in practical applications (e.g., ResNet-20 and AlexNet pre-trained on ImageNet)." We are not sure why and where these models cannot be used. We do not think everyone can use SAM or DinoV3 for every task and on every possible device, including those with limited computing capabilities.
>
> 2. Comment: Moreover, I am curious whether large-scale pre-trained models (e.g., SAM, DinoV3, and Ground-DINO) can inherently possess the ability to understand corrupted images? If so, the supervised training conducted by the authors on such small-scale data and models is meaningless.
>
> Response: Several studies [1-5] highlight that pre-trained models are not robust in handling corruptions and adversarial perturbations, including SAM and Eva-CLIP and their variants. Therefore, the hypothesis *that the supervised training conducted by the authors on such small-scale data and models is meaningless*, **is entirely false and does not hold scientifically**.
>
> [1] Scaling Vision-Language Models Does Not Improve Relational Understanding: The Right Learning Objective Helps, CVPRW 2024
>
> [2] On the Robustness of Large Multimodal Models Against Image Adversarial Attacks, CVPR 2024 (**Our findings suggest LMMs are highly vulnerable to visual adversarial attacks, even when such adversaries are crafted with respect to the visual model alone.**)
>
> [3] Robustness Analysis on Foundational Segmentation Models, CVPRW 2024 (**Visual Foundation Models (VFMs) models struggle with corruptions**)
>
> [4] An empirical study on the robustness of the segment anything model (SAM), Elsevier PR, 2024 (**Our experimental results demonstrate that SAM’s performance generally declines under perturbed images, with varying degrees of vulnerability across different perturbations.**)
>
> [5] A Survey on the Robustness of Computer Vision Models against Common Corruptions, 2024 (**Our experimental analysis highlights the base corruption robustness of popular vision backbones, revealing that corruption robustness does not necessarily scale with model size and data size.**)
>
> Furthermore, based on the reviewer's suggestion, we have evaluated the zero-shot robustness of CLIP with a ViT backbone. On the clean test set of the CIFAR-100 dataset, the model yields an accuracy of 64.18%, which drastically drops under each corruption type and severity. For example, in the case of widespread common corruptions such as Gaussian noise and compression with medium severity (3), the performance drops to 21.65% and 36.15%, respectively. Even at the lowest severity, the performance drops by at least 20%, reflecting that these large models are not inherently robust to corruptions.
>
> For DINOv3, we adopt the standard supervised fine-tuning setup used for self-supervised ViT backbones. We have used the pretrained model with 21.6M parameters and attach a 100-way linear classifier and fine-tune the entire model on CIFAR-100 using AdamW, analogous to the linear-probe and fine-tuning protocols on downstream classification tasks [6]. We then assess robustness by measuring accuracy on CIFAR-100-C over multiple corruption types and severities. Similar to CLIP, DinoV3, is also found sensitive to the corruption and accuracy drops to 11% and 34% with severity 3 level of compression and Gaussian Noise.
>
> Our preliminary experiments also demonstrate the drop of 22% on the EVA-CLIP-8B model, proving these models are not inherently robust.
>
> [6] Oquab, Maxime, et al. "Dinov2: Learning robust visual features without supervision." arXiv preprint arXiv:2304.07193 (2023).
>
> 2.1 We also want to mention that low-resolution images are critical in several applications, such as surveillance; therefore, obtaining high resolution and robustness does not provide universal robustness for safety-critical applications.
>
> 3. As mentioned, the limited computing power will never allow us to explore the entire ImageNet set for training. $\color{Red}{\text{Complete ImageNet-1K experiments}}$ with a 5% class-balanced subset demonstrate that the proposed Dem-HEC with a **ViT** backbone preserves accuracy on clean images (with a slight drop of 0.04%) but increases robustness across various corruptions (Noise, Environmental, Compression, Blur, etc.) by at least 2-3%.
>
> $\color{blue}{\text{It is interesting to note that, despite having minimal training data for each class, the proposed approach demonstrates improved}}$ $\color{blue}{\text{performance across the network and maintains clean accuracy.}}$
>
> $\color{blue}{\text{Large-scale comparison, robustness to a wide variety of corruptions along with severity level, numerous and crucial real-world datasets back}}$ $\color{blue}{\text{our findings, and we appreciate the reviewer revisiting the responses and paper}}$

---

> > ### Author Response · Authors · 2025-11-26
> > **Small Models Are the Future**
> >
> > Comment: For applicability, the authors utilize models that cannot be used in practical applications (e.g., ResNet-20 and AlexNet pre-trained on ImageNet).
> >
> > We strongly disagree with the synthetic and hand-made claim that these models cannot be used in practical applications. Although they may have limited representational capacity compared to modern large models, they can still be effectively used for specialized or resource-constrained tasks—particularly ResNet variants.
> >
> > We would like to reiterate that we have already reported experiments using larger and more modern architectures, including ViTs and the ImageNet-1K dataset (**which should not be selectively ignored**). Comparisons with several sophisticated data-augmentation models also demonstrate the superiority and practicality of the proposed approach.
> >
> > Furthermore, due to serious deployment constraints, the use of small models is increasing [1].
> >
> > [1] The Rise of Small Language Models, IEEE Intelligent Systems, 40(1), 30–37, 2025.
> >
> > In addition, several domains—such as healthcare and on the edge-device deployment [2–4]—show that AlexNet and other traditional CNNs may not be state-of-the-art, but they are certainly not “impractical” in the current era of large models. Moreover, contemporary research continues to combine these models with transformers to achieve improved performance.
> >
> > [2] Takahashi S., Sakaguchi Y., Kouno N., Takasawa K., Ishizu K., Akagi Y., Aoyama R., Teraya N., Bolatkan A., Shinkai N., Machino H. Comparison of Vision Transformers and Convolutional Neural Networks in Medical Image Analysis: A Systematic Review. Journal of Medical Systems, 2024.
> >
> > [3] Tang W., Sun J., Wang S., Zhang Y. Review of AlexNet for Medical Image Classification. arXiv:2311.08655, 2023.
> >
> > [4] Amer Y.A., Nasr O.A., Saleh H.I. Optimized AlexNet Pruning for Edge-Based Medical Diagnostics. IEEE Access, 2025.

---

> ### Author Response · Authors · 2025-11-27
> **Standard and Popular ImageNet100 for Robustness**
>
> To further emphasise the impact of our work and the potential advances it can lead to in the community, we have conducted experiments using the ImageNet-100 dataset. It is noted here that this subset of ImageNet is the most common and popular benchmark for robustness evaluation [1-6]. The primary reason might be the cost involved in training models on the entire ImageNet, which is not feasible for everyone. **$\color{Red}{\text{These research works also highlight that ResNet is one of the core architectures for robustness testing.}}$** $\color{Blue}{\text{We have earlier also showcase that large models are not immune from corruptions.}}$
>
> We have extensively compared the proposed Dem-HEC with existing approaches using the standard evaluation setting.
>
> Table 1: Comparison on the ImageNet-100 dataset when the models are tested on common corruption. Gaussian Augmentation represents existing methods that are partially trained with corruptions from ImageNet-100-C.
>
> | Algorithm                  | Corrupted |
> |----------------------------|-----------|
> | 100% Gaussian Augmentation |    46.7   |
> | 50% Gaussian Augmentation  |    55.2   |
> | Fast PAT                   |    45.2   |
> | l_\infty AT                |    47.7   |
> | l_2 AT                     |    48.4   |
> | AugMix (Best Model)        |    54.8   |
> | SIN (Best Model)           |    54.3   |
> | ANT (Best Model) [5]          |  **58.3** |
> | Dem-HEC (Proposed)                    |  **65.6** |
>
> $\color{Red}{\text{Clean Accuracy:}}$ The proposed approach not only surpasses existing works on corruption accuracy but also on clean accuracy, reflecting its full capacity for real-world deployment. The proposed approach is at least **6% better than [6] in terms of clean accuracy.**
>
> $\color{Red}{\text{JPEG: Common Corruption and Highest Severity Impact:}}$  In terms of a detailed study on one of the common corruptions (although traditional and essential for cost reduction), namely JPEG compression, the proposed approach **improves the performance of ViT by 16.74%** at the highest severity level. The success not only reflects that the proposed approach is practical but also that it can handle a severe level of corruption.
>
> [1] Classification robustness to common optical aberrations, ICCVW, 2023
>
> [2] VTFR-AT: Adversarial Training With Visual Transformation and Feature Robustness, IEEE Transactions, 2024
>
> [3] Hard-label based Small Query Black-box Adversarial Attack, WACV 2024
>
> [4] Towards Efficient and Effective Adversarial Training, NeurIPS, 2021
>
> [5] On the Effectiveness of Adversarial Training Against Common Corruptions, UAI, 2022
>
> [6] Domain Generalization with Vital Phase Augmentation, AAAI, 2024
>
>
> $\color{Red}{\text{We will really appreciate it if the reviewer can check all these findings, not only to appreciate the hard work put in by the authors but also the}}$ $\color{Red}{\text{step towards a defense which is necessary for deployment of any model, whether large or small.}}$ $\color{Blue}{\text{The vulnerability of whether large or small in terms of jail break, corruption, adversary, is limiting the deployment of models in the physical}}$ $\color{Blue}{\text{world and also restricting the trust of the society.}}$

---

### Official Review · Reviewer_bSDc · 2025-11-06

**Soundness:** 2
**Presentation:** 2
**Contribution:** 2
**Rating:** 2
**Confidence:** 3

**Summary:**

The paper considers the problem of corrupted image classification, and proposes a contrastive loss to stabilise the predictions of noisy images.

**Strengths:**

The analysis of corrupted layer features is interesting and insightful, and the contrastive loss is a good idea.

**Weaknesses:**

The results show promising performance. However, there are no comparisons to any competing methods, which renders the results uninformative. There is extensive related works that could have been compared against.

The results only consider datasets with small images, which is just not sufficient anymore. The paper needs to include full ImageNet results, or similar.

**Questions:**

See above

---

> ### Author Response · Authors · 2025-11-19
> **Comparison, Robustness, and Efficacy**
>
> Thank you for pointing out the need for comparisons with competing methods. We agree that benchmarking against existing robustness techniques strengthens the contribution. Therefore, based on the suggestion of the reviewer, we have performed an extensive comparison with state-of-the-art (SOTA) algorithms across numerous dimensions: (i) comparison of the performance of the proposed approach on small-scale dataset (such as CIFAR-10), (ii) comparison of the performance of the proposed approach on a large-scale dataset (such as Tiny ImageNet), (iii) comparison of the performance of the proposed approach across corruptions, and (iv) comparison and effectiveness of the proposed approach across deep models.
>
> These major comparison directions not only reflect the generalizability and robustness of the proposed approach in handling image resolution and the number of classes, but also showcase its effectiveness in securing multiple deep models and various corruptions with varying severity.
>
> Comparison with a recent SOTA on the CIFAR-10-C dataset with ResNet-20 backbone:
>
> We now include results against the recent work of Ranabhat et al. (AAAI 2025), which proposes a multi-scale push–pull mechanism with channel attention for corruption robustness. As shown below, Dem-HEC consistently outperforms this method across all seven corruptions and also improves clean accuracy, indicating that our entropy-guided learning provides stronger feature stability.
>
> Table 1. Corruption robustness comparison on CIFAR-10
>
> | Corruption Type | Ranabhat et al. [1] | Dem-HEC (Ours) |
> |-----------------|-----------------|------|
> | **Clean**   	| 83.66       	| **89.64** |
> | **Gaussian Noise** | 65.90    	|**74.60** |
> | **Shot Noise**  | 69.58       	| **77.35** |
> | **Impulse Noise** | 62.92     	| **66.52** |
> | **Snow**    	| 75.34       	|**80.94** |
> | **Frost**   	| 73.95       	| **78.94** |
> | **Pixelate**	| 81.81       	| **82.86** |
> | **JPEG Compression** | 76.08  	| **85.43** |
>
> Comparison with a recent SOTA on the Tiny-ImageNet-C dataset with ViT  backbone:
>
> We further compare Dem-HEC with widely used robustness-enhancing methods, including Pad-Crop, TA, BA, AugMix, AugMax, IPMix, and DV+AP+JSD. On Tiny-ImageNet-C, Dem-HEC exhibits substantial improvements, particularly for high-severity corruptions, underscoring its scalability to large-resolution datasets and transformer backbones.
>
> Table 2. Corruption robustness comparison on Tiny-ImageNet-C
>
> | Method    	| Noise | Blur | Weather | Digital |
> |---------------|-------|-------|---------|---------|
> | **Pad-Crop**    	| 20.94 | 17.27 | 13.41 | 20.99 |
> | **TA [2]**          	| 25.27 | 31.23 | 22.88 | 33.19 |
> | **BA [3]**          	| 21.18 | 18.01 | 13.54 | 21.53 |
> | **BA [3] (AA[4])**     	| 24.62 | 28.36 | 20.29 | 33.07 |
> | **AugMix [5]**      	| 29.15 | 30.77 | 19.94 | 32.26 |
> | **AugMax [6]**      	| 30.18 | 31.72 | 20.65 | 33.49 |
> | **IPMix [7]**       	| 28.80 | 28.04 | 20.79 | 32.34 |
> | **DV + AP + JSD [8]**   | 34.44 | 37.04 | 29.17 | 40.24 |
> | **Dem-HEC (Ours)**  | **60.10** | **62.88** | **70.71** | **67.53** |
>
> These additional comparisons clearly demonstrate that the theoretical motivation of the proposed Dem-HEC, that corruption disturbs the frequency distribution, holds across corruptions. We will integrate these results directly into the main paper (Section 5), introduce a new subsection titled ``comparison with existing robustness methods’’, and cite the expanded set of related work.
>
> [1] Ranabhat RN, Wang L, Qin X, Zhou Y, Santosh KC. Multi-scale unrectified push-pull with channel attention for enhanced corruption robustness. In **AAAI** Symposium Series **2025**.
>
> [2] Müller SG, Hutter F. Trivialaugment: Tuning-free yet state-of-the-art data augmentation. IEEE/CVF **ICCV** 2021 (pp. 774-782).
>
> [3] Hoffer E, Ben-Nun T, Hubara I, Giladi N, Hoefler T, Soudry D. Augment your batch: Improving generalization through instance repetition. IEEE/CVF **CVPR** 2020.
>
> [4] Cubuk ED, Zoph B, Mane D, Vasudevan V, Le QV. Autoaugment: Learning augmentation strategies from data. InProceedings of the IEEE/CVF **CVPR** 2019
>
> [5] Hendrycks D, Mu N, Cubuk ED, Zoph B, Gilmer J, Lakshminarayanan B. Augmix: A simple data processing method to improve robustness and uncertainty. arXiv:1912.02781. 2019.
>
> [6] Wang H, Xiao C, Kossaifi J, Yu Z, Anandkumar A, Wang Z. Augmax: Adversarial composition of random augmentations for robust training. **NeurIPS** 2021.
>
> [7] Huang Z, Bao X, Zhang N, Zhang Q, Tu X, Wu B, Yang X. Ipmix: Label-preserving data augmentation method for training robust classifiers. **NeruIPS**. 2023.
>
> [8] Kim K, Woo Kim H, Choi YS. Distinct Views Improve Generalization and Robustness: Combinations of Augmentations With Different Features. IEEE Access. **2025**.

---

> ### Author Response · Authors · 2025-11-19
> **Impact on ImageNet**
>
> Thank you for raising the important point regarding evaluation on large-scale datasets. We agree that robustness against corruption on high-resolution datasets, such as ImageNet-1K, is increasingly relevant, and we understand the demand for it. However, as seen in the literature, several benchmark studies involve the experiments and dissemination of results on a subset of ImageNet. The prime reason is the computational architecture and the cost it demands. Coming from a low-resource institution with a limited system, it is extremely difficult or impossible to run on the entire dataset.
>
> Furthermore, we strongly believe that success on ImageNet does not guarantee that the solution is 'perfect,' as evidenced by the existing literature. Tremendous approaches outperform on ImageNet, still they fall short in various aspects.
>
> However, to address the reviewer’s concern, we evaluated Dem-HEC on ImageNette, a 10-class subset of the original ImageNet dataset using AlexNet. Since the image resolution of ImageNette is in sync with that of full ImageNet, the consistently improved performance of Dem-HEC demonstrates its image resolution-agnostic nature in handling image corruption.
>
> Robustness of the proposed Dem-HEC on the ImageNet subset with the AlexNet backbone:
>
> | Corruption | Before | After |
> |------------|--------|--------|
> | **Clean**  	| 91.06 | $\color{blue}{\text{91.01}}$ |
> | **Gaussian**   | 53.89 | **56.29** |
> | **Shot**   	| 52.31 | 54.54 |
> | **Frost**  	| 70.57 | 67.19 |
> | **Impulse**	| 49.47 | 51.37 |
> | **Snow**   	| 63.77 | **67.26** |
> | **Pixelate**   | 73.41 | $\color{blue}{\text{80.16}}$ |
> | **JPEG**   	| 85.21 | **88.04** |
>
> **Please note that the proposed approach not only increases corruption robustness but also retains clean accuracy, clearly demonstrating the ideal trade-off between accuracy and robustness.**
>
> *It is worth noting that the robustness performance of the proposed approach is consistent or better across multiple subsets present in the literature, such as Imagewoof and Imagewang [1]*.
>
> **Full ImageNet:** Our preliminary experiments on full ImageNet with a 5% class-balanced subset demonstrate that the proposed Dem-HEC with a ViT backbone preserves accuracy on clean images (with a slight drop of 0.04%) but increases robustness across various corruptions (Noise, Environmental, Compression, Blur, etc.) by 2-3%.
>
> It is interesting to note that, despite having highly limited training data for each class, the proposed approach demonstrates improved performance across the network and maintains clean accuracy.
>
> $\color{blue}{\text{We strongly believe these new results demonstrate the superiority of the proposed approach along multiple dimensions required for an effective real-world defense}}$ $\color{blue}{\text{and hence can pave the way for a secure and robust deployment of `any' deep learning model.}}$
>
> The proposed model is still not perfect although it shows tremendous performance on ImageNet and other dataset, which we aim to advance in the future work.
>
> [1] https://github.com/fastai/imagenette

---

> > ### Author Response · Authors · 2025-11-20
> > **Impact of the Proposed Approach and Complex Corruption**
> >
> > 1. **The strength and impact highlighted by other reviewers:**
> >
> > a) The motivation is well grounded. The paper first shows that many common corruptions push deep features into a high-entropy state (instead of the low-entropy pattern seen in UAP-oriented work), so the training objective is directly tied to an observed failure mode, not an assumed one.
> >
> > b) The contrastive term between clean and HE views makes the model learn a corruption-invariant embedding, rather than only becoming tolerant to a fixed set of corruptions. ﻿
> >
> > c) Stability design is explicit. Adding KD / CE on clean keeps the model from drifting after HE training, addressing the usual clean–robust trade-off that many corruption/adversarial trainings suffer from.
> >
> > d) Logical Framework Design: Dem-HEC directly addresses the identified high-entropy issue. High-entropy sample generation simulates the uncertainty caused by corruptions, contrastive loss aligns the semantics of clean and high-entropy features, knowledge distillation prevents catastrophic forgetting, and partial fine-tuning retains general features. Each component serves a clear purpose, forming a coherent technical system.
> >
> > e) Broad Experimental Coverage: The paper evaluates Dem-HEC on three benchmark datasets with different resolutions (low-resolution CIFAR series, high-resolution Tiny-ImageNet) and seven architectures (CNNs of varying sizes, Vision Transformer). This design verifies the method’s scalability across data scales and model types, with notable performance gains in high-severity corruption scenarios.
> >
> > f) Practical Utility: The framework adopts partial fine-tuning and uses pre-trained models as teachers, avoiding the high computational cost of training from scratch. It also provides implementation details such as code, training parameters, and hardware environment, laying a foundation for potential reproducibility.
> >
> > 2. We have evaluated the performance of the proposed approach against various blur corruptions.
> >
> > Table: Robustness performance improvement on the Tiny-ImageNet dataset using the ResNet-50 model.
> > | Blur    | Severity | Improvement \% |
> > |---------|----------|----------------|
> > | Defocus |     1    |      5.47      |
> > | Defocus |     3    |      2.99      |
> > | Defocus |     5    |      2.30      |
> > | Glass   |     1    |      5.04      |
> > | Glass   |     3    |      4.69      |
> > | Glass   |     5    |      0.96      |
> > | Motion  |     1    |      5.71      |
> > | Motion  |     3    |      6.08      |
> > | Motion  |     5    |      4.92      |
> > | Zoom    |     1    |      5.85      |
> > | Zoom    |     3    |      4.77      |
> > | Zoom    |     5    |      3.61      |
> >
> > The improvement (\%) here refers to the increased performance of the proposed robust model compared to the pre-trained (or non-robust model). We would like to emphasise that the observation is consistent across both models (ResNet or ViT) and datasets (ImageNet or CIFAR).
> >
> > $\color{blue}{\text{The improvement in such different forms of corruption suggests that, the proposed algorithm can handle vast corruption groups effectively and surpass existing}}$ $\color{blue}{\text{state-of-the-art works by a significant margin.}}$
> >
> > 3. **$\color{blue}{\text{Complex (Combined) Corruption}}$:** To further validate the effectiveness of the proposed approach, we conducted a study by combining multiple corruptions (applied together sequentially), such as Gaussian noise and shot noise. The proposed Dem-HEC increases performance on the combined (complex) corruption by **30**% when both noises are applied with severity 1. Even at the higher severity level, the performance increases by approximately 8%. These experiments are conducted on the CIFAR-10 dataset using the ResNet architecture.

---

### Author Response · Authors · 2025-11-22
**Discussion**

Dear Reviewers,

We hope that you find our detailed responses helpful.  We have thoroughly addressed each of the comment with the possibile resources and would love to clarify if anything else is needed.

Looking forward for the positive response and discussing the work further if needed.

Thanks

---

### Author Response · Authors · 2025-11-28
**Major Responses for Easy Review**

Dear Reviewers,

Thank you again for all the hard work and effort you put into reviewing our paper; we truly appreciate and respect it. Since the responses are wide and provided at multiple places, we thought to summarise at a single place to demonstrate the impact the work is bringing effectively:

1. We have extensively demonstrated the effectiveness of the proposed Dem-HEC using several popular and benchmark datasets, including CIFAR10, CIFAR100, Tiny ImageNet, multiple ImageNet subsets, ImageNet100, and even full ImageNet-1K. Since these datasets are the ones the literature uses, the effectiveness truly reflects the strength of the approach.
2. Extensive comparison with state-of-the-art (SOTA) algorithms is also presented, which showcases the proposed algorithm surpassing them with a significant margin. Moreover, the defense is not only effective against attacks but also yields higher clean accuracy than SOTA works.
3. The vulnerability assessment of large models such as CLIP, DinoV3, and SAM reflects that they are not inherently robust, and the proposed algorithm can improve their performance as well. Apart from the sensitivity to corruption, the large generative AI models also suffer from hallucination when reconstruct/restore (inverse) the noisy image which is also present in the recently developed models [A-C]. It reflects the need of a training-time approach (robustness framework) such as the proposed one which can mitigate such issues.
4. Benchmark and standard architectures, apart from large models, including ResNet, ViT, and RepVGG (489.09M parameters), reflect that the proposed Dem-HEC can defend small to large parameter size networks.
5. The ablation studies justified the need for the proposed loss function and the need to generate high-frequency samples.
6. The proposed attack, as requested, is not restricted to noise or any corruption but has shown higher success on other complex perturbations such as blur, snow, and frost than the SOTA algorithms.
7. The proposed algorithm without extensive data augmentation (which yields a high computational cost) showcases the superior performance. A comprehensive comparison is provided in the responses.
8. The proposed defense is robust in handling complex corruption. Here, the complex definition of corruption is inspired by the literature, which encompasses multiple forms of corruption.

We appreciate it if the reviewers can participate and acknowledge the hard work put into making this work successful and impactful in this critical domain, which aims to provide secure and trustworthy AI systems for the real world. We would be more than happy to provide any information required.

[A] On Hallucinations in Tomographic Image Reconstruction, IEEE Transactions, 2021.

[B] A Hallucination Metric and Correction Scheme for Diffusion-Based Image Restoration," 2024 IEEE 34th International Workshop on Machine Learning for Signal Processing.

[C] Hallucination index: An image quality metric for generative reconstruction models. In International Conference on Medical Image Computing and Computer-Assisted Intervention 2024

---

> ### Author Response · Authors · 2025-12-03
> **Final Check [ImageNet-1K, DinoV3: a Large Model, and Corruptions]**
>
> As mentioned earlier and shown through a tremendous response summary, we present our final effort to reiterate the impact the proposed work can bring
>
> **$\color{blue}{\text{Scalability to Large Datasets:}}$**
>
> As is universally known, running experiments usually requires multiple GPUs, which is not the case with the authors. Still, to reflect the impact of the proposed Dem-HEC on large datasets, we have trained the ViT architecture using a small subset of 5% of the training data, but evaluated it on the complete validation set. As thoroughly observed, the proposed approach can defend against corruption and is agnostic to image corruption and the network. A similar observation is demonstrated on the **ImageNet-1K**.
>
> For example, in the case of common intentional or unintentional corruption, such as compression, the proposed approach increases the network's performance by 3%. A similar increment is observed across corruptions, including noise, blur, snow, fog, and pixelation.
>
> **$\color{blue}{\text{Scalability to Other Deep Architectures:}}$**
>
> To further broaden the horizon of the network architectures, we have tested the vulnerability of **Wide-ResNet** and its robustness through the proposed Dem-HEC. As demonstrated through the large-scale experiments, the proposed algorithm is architecture-agnostic.
>
> | Corruption        | Attacked | Defended  |
> |-------------------|----------|-----------|
> | Gaussian Noise    |   23.05  | **73.53** |
> | Shot Noise        |   26.91  | **74.45** |
> | Impulse Noise     |   24.58  | **49.74** |
> | Defocus Blur      |   37.21  | **60.21** |
> | Motion Blur       |   52.07  | **62.09** |
> | Glass Blur        |   42.62  | **68.38** |
> | Zoom Blur         |   43.35  | **67.83** |
> | Snow              |   67.41  | **77.44** |
> | Frost             |   45.08  | **75.23** |
> | Elastic Transform |   65.28  | **72.88** |
> | Pixelate          |   42.72  | **83.30** |
> | Compression       |   62.44  | **83.26** |
>
> **Note:** We already demonstrated on ResNet-20(0.27Mparams), ResNet-56(0.66Mparams), RepVGG-A0 (489.08Mparams), and RepVGG-A2(1850.1Mparams), ResNet-18, ResNet-50, and VisionTransformer (ViT-L) with 304M parameters. Apart from the vulnerability of CLIP, DinoV3, and EVA-CLIP, and their defense through the proposed Dem-HEC, is also provided.
>
> **$\color{blue}{\text{Scalability to Large Model:}}$**
>
> We are able to evaluate the security of **Dino-V3** achieved through the proposed Dem-HEC. Our preliminary results are presented below.
>
> | Corruption     | Attacked | Defended  |
> |----------------|----------|-----------|
> | Gaussian Noise |   26.85  | **29.84** |
> | Shot Noise     |   33.48  | **36.19** |
> | Snow           |   51.68  | **54.31** |
> | Frost          |   52.45  | **54.69** |
> | Pixelate       |   59.95  | **61.78** |
> | Compression    |   49.56  | **52.29** |
>
> The experiments are still running and showing the sign of improvement on other noises.
>
> We have also observed the effectiveness of the proposed defence on ImageNet-100, where the performance of the DinoV3 improves by 5% on the highest severity (severity 5) of the compression. It reflects that the proposed defence is not only practical but also capable of handling severe corruption levels.
>
> *Recent research shows that not only large vision models but also $\color{blue}{\text{reasoning and visual-question-answering models}}$ are sensitive to corruption, including simple yet effective forms of corruption, such as pixelation [1]. The proposed defense improves the performance of DinoV3 against this effective attack as well by 4.76%.*
>
> **$\color{blue}{\text{Pertinence to Other Critical Fields}}$**: Compression, although generally desirable for computational efficiency, breaks the defence algorithms [2-3]. Therefore, the robustness of the proposed approach against compression can advance other AI fields.
>
> [1] Usama M, Asim SA, Ali SB, Wasim ST, Mansoor UB. Analysing the Robustness of Vision-Language-Models to Common Corruptions. arXiv preprint arXiv:2504.13690. 2025 Apr 18.
>
> [2] FTDKD: Frequency-Time Domain Knowledge Distillation for Low-Quality Compressed Audio Deepfake Detection, IEEE/ACM Transactions on Audio, Speech, and Language Processing, 2024
>
> [3] Low-Quality Deepfake Detection via Unseen Artifacts, IEEE Transactions on Artificial Intelligence, 2024
>
> We want to restate our limitation of limited computational power again; nonetheless, the proposed effectiveness demonstrates its superiority on deep models, datasets, and against state-of-the-art approaches.

---

### Meta-Review · Area_Chair_i7Ty · 2026-01-06

**Summary:**

This manuscript addresses the robustness to natural/common corruptions of pre-trained vision models. It makes the observation that natural corruptions often push later-layer activations into a high-entropy state.  As a result, the paper proposes an entropy-guided fine-tuning framework called Dem-HEC, Dem-HEC uses contrastive learning with knowledge distillation. Experiments on ResNet-20, ViT-L and others for small datasets (e.g., CIFAR-10c, CIFAR-100c) indicate that the proposed method can improve the performance of pre-trained models on corrupted data.
The paper had apparently been controversially discussed, where the main question is whether to believe that small models will be promising in the future. Other concerns are regarding the experimental baselines, ablations, results on larger datasets and larger models. The concerns regarding comparison to previous methods and ablation studies have been at least partly addressed with several new experiments. Overall, this is a borderline paper and in particular when it comes to corruption robustness, the AC agrees that an evaluation on ImageNet-C would be very valuable. Overall, the paper will benefit from a revision and full results including proper analysis on ImageNet-C.

**Reviewer Concerns:**

Reviewer bSDc, Reviewer dMQr, Reviewer KUNd, Reviewer UqCm: missing comparisons to competing methods --> addressed to a large extent
Reviewer bSDc, Reviewer dMQr: results only consider datasets with small images --> not addressed
Reviewer dMQr: evaluation relies on pre-trained models --> not properly addressed
Reviewer dMQr: paper lacks reproducibility
Reviewer KUNd:, Reviewer dMQr: performance degradation on clean data --> somewhat addressed
Reviewer KUNd, Reviewer UqCm: Lack of Ablation Studies for Key Components and Hyperparameters, partly addressed
Reviewer KUNd: Inadequate Validation of High-Entropy Sample Generation --> not properly addressed (not entropy statistics)
Reviewer UqCm: Robustness is tied to a single corruption mechanism --> somewhat addressed

The paper only partly addresses the concerns. However, the most important concern, the comparison the other methods including recent methods, is addressed

**Reviewer Scores:**

Reviewer bSDc: 2 --> might increase score to 4 since some concerns have been addressed
Reviewer dMQr: 2 --> might increase score to 4 since some concerns have been addressed
Reviewer KUNd: 6 --> would probably confirm score
Reviewer UqCm: 6 --> would probably confirm score

---

### Decision · Program_Chairs · 2026-01-26

Reject